# SIMULATING TRAINING DYNAMICS TO RECONSTRUCT TRAINING DATA FROM DEEP NEURAL NETWORKS

**Hanling Tian, Yuhang Liu, Mingzhen He, Zhengbao He, Zhehao Huang, Ruikai Yang, Xiaolin Huang** [*]
Institute of Image Processing and Pattern Recognition, Shanghai Jiao Tong University
{hanlingtian, yuhangliu, mingzhen_he, lstefanie, kinght_h, ruikai.yang, xiaolinhuang}@sjtu.edu.cn

## ABSTRACT

Whether deep neural networks (DNNs) memorize the training data is a fundamental open question in understanding deep learning. A direct way to verify the memorization of DNNs is to reconstruct training data from DNNs' parameters. Since parameters are gradually determined by data throughout training, characterizing training dynamics is important for reconstruction. Pioneering works rely on the linear training dynamics of shallow NNs with large widths, but cannot be extended to more practical DNNs which have non-linear dynamics. We propose **Simu**lation of training **Dy**namics (SimuDy) to reconstruct training data from DNNs. Specifically, we simulate the training dynamics by training the model from the initial parameters with a dummy dataset, then optimize this dummy dataset so that the simulated dynamics reach the same final parameters as the true dynamics. By incorporating dummy parameters in the simulated dynamics, SimuDy effectively describes non-linear training dynamics. Experiments demonstrate that SimuDy significantly outperforms previous approaches when handling non-linear training dynamics, and for the first time, most training samples can be reconstructed from a trained ResNet's parameters. Our code is available at
https://github.com/BlueBlood6/SimuDy.

## 1 INTRODUCTION

Deep neural networks (DNNs) have shown remarkable performance and generalization across various tasks (Hinton et al., 2012; Devlin et al., 2019; Wu et al., 2022), due to their powerful learning capabilities. An intriguing question is what DNNs have learned or even memorized. This memorization is closely related to the generalization capability (Feldman, 2020; Feldman & Zhang, 2020), and also raises concerns about the potential leakage of private information within training data (Shokri et al., 2017; Carlini et al., 2019; 2021; 2023).

During training, DNNs' parameters are progressively adjusted by the training data. Roughly, when the learning algorithm and the initial parameters are given, the final parameters are determined by the training data. This mapping indicates that the training data may be *memorized* within the DNNs' parameters. A direct way to support this guess is to reconstruct the training dataset from parameters. However, this reconstruction task is challenging due to the complexity of the training process, which involves disentangling parameter changes at each step and decoupling each sample's contribution from the cumulative sum of gradients. Gradient inversion attack (Zhu et al., 2019) could recover data from the gradient of a single step on dozens of images. In fact, the two tasks are related but essentially different. Reconstructing training data from parameters are more difficult and necessitates a deep understanding of training dynamics, as it involves recovering data from the cumulative effects of multiple training steps, rather than simply considering a single step's gradient. In other words, gradient inversion attack is the simplest case of reconstruction from parameters when the training process contains only one step.

To reconstruct training data from parameters, we need to deeply study the training dynamics. The simplest training dynamics are linear, meaning the directions of gradients for each data point remain consistent, as seen in cases for shallow NNs with infinite widths. For an approximately linear scenario: training a three-layer multilayer perceptron (MLP) model with 1000 neurons in each layer,

---

[*]Corresponding author

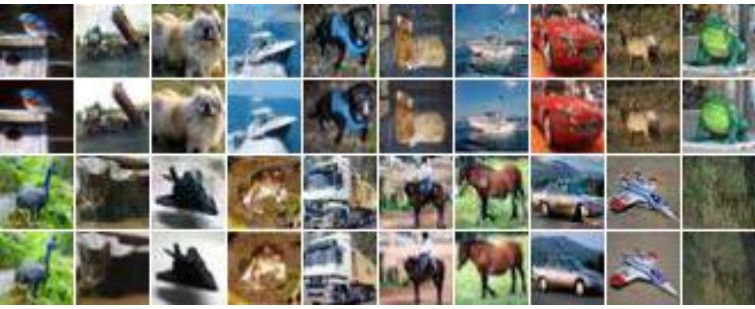

Figure 1: Reconstructed training samples from a multi-class ResNet using our SimuDy(rows 1, 3), and the corresponding nearest neighbors from the training set (rows 2, 4).

Haim et al. (2022) pioneer a reconstruction method based on the Karush–Kuhn–Tucker (KKT) conditions of a certain max-margin problem (Lyu & Li, 2020; Ji & Telgarsky, 2020) where gradient flow converges in direction. This method successfully reconstructed about 50 out of 500 CIFAR-10 training images from the MLP's final parameters. Similarly, Loo et al. (2024) reconstruct the training data from NNs in the neural tangent kernel (NTK) (Jacot et al., 2018) regime, where the training dynamics are also linear.

However, the reconstruction should not be limited to linear training dynamics, which are rarely encountered in real-world applications. The key to dataset reconstruction is accurately describing the gradients at each step since parameters and data are linked via these gradients. In linear dynamics, the gradient directions remain consistent, so the training dynamics can be effectively characterized by even only the final parameters. While for non-linear dynamics, Loo et al. (2024) make a step by considering the linear combination of gradients from initial parameters and final parameters, achieving better reconstructions from MLPs than Haim et al. (2022). Nevertheless, using the sum of gradients from initial and final static parameters to characterize dynamics, is still too simple for popular and practical cases, i.e., DNNs such as ResNet (He et al., 2016) trained with mini-batch Stochastic Gradient Descent (SGD) (Bottou, 2010).

In this paper, we propose a novel framework called "**Simu**lation of training **Dy**namics" (SimuDy) to reconstruct the training data from parameters of trained DNNs. As the name suggested, our method constructs the simulated training dynamics by training the model from the initial parameters with a dummy dataset initialized with random Gaussian noise. Then, we optimize the dummy dataset to make the simulated training dynamics converge to the same final parameters as the original training dynamics, with the dummy dataset progressively refined towards the real dataset. The detailed characterization of the non-linear training dynamics aids in data reconstruction from DNNs. Our results demonstrate that SimuDy can reconstruct training data with high quality from a ResNet trained using mini-batch SGD, where previous methods have failed. And an example using SimuDy is shown in Fig. 1. For the first time, a large portion of the training data can be reconstructed from a trained ResNet's parameters. Also, the reconstruction performance remains excellent even with unknown hyper-parameters of model training. Our contributions could be summarized as follows:

- We propose SimuDy to reconstruct training data from parameters of trained DNNs, which are more practical than MLPs in realistic applications.
- Our method illustrates the importance of characterizing training dynamics for dataset reconstruction, which would provide insights for memorization of DNNs.
- Extensive experiments show that SimuDy outperforms previous methods when dealing with non-linear training dynamics. Additionally, we demonstrate our method's effectiveness and robustness with unknown hyper-parameter settings of model training.

## 2 RELATED WORK

**Dataset Reconstruction.** The pioneering work (Haim et al., 2022) has shown that training data can be reconstructed from trained homogeneous neural network parameters. They note that homogeneous neural networks converge in direction to the solution of a max-margin problem (Lyu

& Li, 2020; Ji & Telgarsky, 2020). By optimizing dummy images and dual parameters to match Karush–Kuhn–Tucker (KKT) conditions of the max-margin problem, training data is reconstructed. Then, Buzaglo et al. (2024) extend the reconstruction scheme of Haim et al. (2022) to a multi-class setting and MLPs with a convolutional layer in the front. Loo et al. (2024) provably reconstruct the entire training set for networks in the neural tangent kernel (NTK) (Jacot et al., 2018) regime where the directions of gradients for each data point does not change during training, i.e., the training dynamics are linear. Toward non-linear dynamics, they take a step to linearly combine the initial and final gradients of data to represent parameter changes and get better reconstructions. Aforementioned methods mainly focus on shallow and wide MLPs where the training dynamics are linear. Besides MLPs, we can reconstruct training data from practical DNNs with non-linear dynamics.

**Gradient Inversion Attack.** Gradient inversion attack aims to reconstruct training data from the batch gradient and the model's parameters. DLG (Zhu et al., 2019) is the pioneering optimization-based method that minimizes the distance between the ground-truth gradients and the dummy gradients of a batch of dummy samples. iDLG (Zhao et al., 2020) derives the ground-truth label from the gradient of the fully connected layer to improve the attack. Geiping et al. (2020) change previous Euclidean distance loss to cosine similarity loss and add total variation (Rudin et al., 1992) as the image prior to reconstruct higher resolution images from ResNet (He et al., 2016). With strong batch normalization statistics, Yin et al. (2021) propose group consistency regularization to recover partial images at a larger batch size even up to 48. Recently, the following works (Jeon et al., 2021; Li et al., 2022; Fang et al., 2023; Zhang et al., 2023) leverage pre-trained generative models to better utilize image prior and improve the quality of reconstructed images. Dataset reconstruction is related to the gradient inversion attack, as both aim to recover images from different forms of gradients. However, dataset reconstruction presents a greater challenge because the gradients come from a dynamic process, whereas gradient inversion attack only considers a single step's gradient.

## 3 SIMULATING TRAINING DYNAMICS

**Task definition.** Given a neural network $f_{\boldsymbol{\theta}}$ with the initial parameters $\boldsymbol{\theta}_0$ and the final parameters $\boldsymbol{\theta}_f$ for multi-class image classification tasks, dataset reconstruction aims to invert the parameters to the training dataset $\mathcal{D}$. Considering the learning algorithm $\mathcal{A}$, which is characterized by the hyper-parameters $H$, the training process maps $\mathcal{D}$ to $\boldsymbol{\theta}_f$: $\boldsymbol{\theta}_f = \mathcal{A}_H(\mathcal{D}; \boldsymbol{\theta}_0)$. In this point of view, the dataset reconstruction is an inverse problem $\mathcal{D} = \mathcal{A}_H^{-1}(\boldsymbol{\theta}_f; \boldsymbol{\theta}_0)$.

**Revisiting training.** The inverse problem $\mathcal{D} = \mathcal{A}_H^{-1}(\boldsymbol{\theta}_f, \boldsymbol{\theta}_0)$ is over-determined in the context of deep learning because a DNN is usually over-parameterized, i.e., the dimension of $\boldsymbol{\theta}_f$ is significantly higher than that of the entire dataset, providing sufficient supervision to reconstruct the training data. However, the data information is embedded into the parameters by iteratively updating the network parameters in the form of gradients. Therefore, to solve the inverse problem, we need to investigate the training process, which can be described by the following differential equation,

$$\frac{\mathrm{d}\boldsymbol{\theta}(t)}{\mathrm{d}t} = -\nabla_{\boldsymbol{\theta}}\mathcal{L}(\mathcal{D}, \boldsymbol{\theta}(t)), \tag{1}$$

with the constraints $\boldsymbol{\theta}(0) = \boldsymbol{\theta}_0$ and $\boldsymbol{\theta}(T) = \boldsymbol{\theta}_f$. In practical scenarios, this continuous process is discretized, enabling the parameter updates to be performed in a series of discrete steps, which makes it feasible to implement. Specifically, one of the most representative methods is SGD, where parameters are updated iteratively by moving against the gradient of the loss function, calculated on mini-batches of the data:

$$\boldsymbol{\theta}_0 - \boldsymbol{\theta}_f = \sum_{k=1}^{T}\sum_{j=1}^{N}\left(\eta \cdot \sum_{\boldsymbol{x},y \in \mathcal{B}_{k,j}} \frac{1}{|\mathcal{B}_{k,j}|}\nabla_{\boldsymbol{\theta}}\ell\left(f_{\boldsymbol{\theta}_{k,j}}(\boldsymbol{x}), y\right)\right), \tag{2}$$

$$\boldsymbol{\theta}_{k,j+1} = \boldsymbol{\theta}_{k,j} - \eta \cdot \sum_{\boldsymbol{x},y \in \mathcal{B}_{k,j}} \frac{1}{|\mathcal{B}_{k,j}|}\nabla_{\boldsymbol{\theta}}\ell\left(f_{\boldsymbol{\theta}_{k,j}}(\boldsymbol{x}), y\right), \quad \boldsymbol{\theta}_{k+1,1} = \boldsymbol{\theta}_{k,N}, \tag{3}$$

where $T$ is the total number of training epochs, $N$ is the total number of batches, $\ell$ is the loss function, $\mathcal{B}_{k,j}$ is the $j$-th batch of the $k$-th epoch, $|\mathcal{B}_{k,j}|$ is the batch size of $\mathcal{B}_{k,j}$, $\boldsymbol{\theta}_{k,j}$ is the model parameters of $k$-th epoch after $j$-th batch, $(\boldsymbol{x}, y)$ is a data point in $\mathcal{B}_{k,j}$, and $\eta$ is the learning rate.

**Linear training dynamics.** The simplest case of dataset reconstruction is that the training dynamics are linear or approximately linear. In this case, the direction of gradients $\nabla_{\boldsymbol{\theta}}$ keeps almost the same

---

**Algorithm 1** Reconstructing training data using SimuDy.

---

**Input:** Network function $f_{\boldsymbol{\theta}}$, initial parameters $\boldsymbol{\theta}_0$, final parameters $\boldsymbol{\theta}_f$, dataset size $n$, training learning rate $\eta$, training steps $T$, batch size $|\mathcal{B}|$, dissimilarity function $\boldsymbol{d}(\cdot, \cdot)$, optimizer `Optim`;
**Output:** Reconstructed images via SimuDy;
 1: Initialize dummy images $\hat{\boldsymbol{x}}$ with random noise
 2: Assign labels to images randomly, ensuring an equal number of labels for each class
 3: Randomly divide the dataset into batches of size $|\mathcal{B}|$, and the number of batches is $N$
 4: $\hat{\boldsymbol{\theta}}_{1,1} \leftarrow \boldsymbol{\theta}_0$                                    ▷ Begin with the initial parameters
 5: **repeat**
 6:     **for** $k = 1$ **to** $T$ **do**                       ▷ Simulate the training process for $T$ epochs
 7:         **for** $j = 1$ **to** $N$ **do**                   ▷ Perform $N$ updates for each epoch
 8:              $\boldsymbol{g}_{k,j} = \sum_{\hat{\boldsymbol{x}}, \hat{y} \in \mathcal{B}_{k,j}} \frac{1}{|\mathcal{B}_{k,j}|} \nabla_{\boldsymbol{\theta}} \ell(f_{\hat{\boldsymbol{\theta}}_{k,j}}(\hat{\boldsymbol{x}}), \hat{y})$     ▷ Compute the gradient for each update
 9:              $\hat{\boldsymbol{\theta}}_{k,j+1} \leftarrow \hat{\boldsymbol{\theta}}_{k,j} - \eta \cdot \texttt{grad\_clip}(\boldsymbol{g}_{k,j})$       ▷ Clip gradients and update parameters
10:         **end for**
11:          $\hat{\boldsymbol{\theta}}_{k+1,1} \leftarrow \hat{\boldsymbol{\theta}}_{k,N+1}$
12:     **end for**
13:     $\hat{\boldsymbol{\theta}}_f \leftarrow \hat{\boldsymbol{\theta}}_{T,N+1}$                             ▷ Get simulated final parameters
14:     $\mathcal{L}_{\text{recon}} = \boldsymbol{d}(\boldsymbol{\theta}_f - \boldsymbol{\theta}_0, \hat{\boldsymbol{\theta}}_f - \boldsymbol{\theta}_0) + \alpha \cdot \mathcal{L}_{\text{TV}}(\boldsymbol{x})$       ▷ Compute reconstruction loss
15:     $\hat{\boldsymbol{x}} \leftarrow \texttt{Optim}(\hat{\boldsymbol{x}}, \partial \mathcal{L}_{\text{recon}} / \partial \hat{\boldsymbol{x}})$                   ▷ Update dummy images
16: **until** the reconstruction loss converges.

---

throughout the training process. Then, by leveraging the direction invariance, Eq. 2 can be written as $\boldsymbol{\theta}_0 - \boldsymbol{\theta}_T = \sum_{i=1}^{|\mathcal{D}|} \lambda_i \cdot \nabla_{\boldsymbol{\theta}} \ell(f_{\boldsymbol{\theta}_0}(\boldsymbol{x}_i), y_i)$, where $\lambda_i$ is the scaling factor addressing the contribution of $\boldsymbol{x}_i$. Thus, the linear combination of gradients computed from the initial or final model parameters can effectively characterize training dynamics. In other words, with linear training dynamics, the dataset reconstruction could be reduced to a problem of extracting images from the gradients of a single step on a batch of data.

**Simulation of training dynamics.** However, the linear training dynamics exist only for some ideal scenarios such as training a shallow MLP with sufficient width (Haim et al., 2022). In practice, the training dynamics are typically non-linear. To reconstruct the dataset from practical DNNs, we should delve into the whole training dynamics traversing from initial parameters $\boldsymbol{\theta}_0$ to final parameters $\boldsymbol{\theta}_f$, rather than solely relying on gradients of these two sets of parameters to characterize the entire training process. Nevertheless, such training dynamics are usually inaccessible, which makes reconstruction for non-linear dynamics seemingly impossible. In this paper, we propose "**Simu**lation of training **Dy**namics" (SimuDy) to reconstruct training data from practical DNNs with non-linear dynamics, i.e., we will construct new training dynamics and optimize the dummy dataset from noise to original data such that the simulated training dynamics converge to the real dynamics. The pseudocode of our proposed SimuDy is specified in Alg. 1. SimuDy introduces additional simulated checkpoints in the training process, which would provide more accurate guidance than only using the initial and final parameters. At a high level, our method can be presented as follows:

$$\min_{\mathcal{X} \in \mathcal{F}} \quad \boldsymbol{d}\left(\mathcal{A}_H(\boldsymbol{\mathcal{X}}; \boldsymbol{\theta}_0), \mathcal{A}_H(\boldsymbol{\mathcal{D}}; \boldsymbol{\theta}_0)\right), \tag{4}$$

where $\mathcal{F}$ denotes the set of possible images, and $\boldsymbol{d}(\cdot, \cdot)$ is the non-negative dissimilarity function.

**Loss design.** Minimizing the dissimilarity between simulated final parameters $\hat{\boldsymbol{\theta}}_f$ and real final parameters $\boldsymbol{\theta}_f$ is to supervise the simulated dynamics to be similar to the underlying dynamics. We choose the cosine similarity loss following Geiping et al. (2020), as the high-dimensional direction of the gradient can carry significant information (Charpiat et al., 2019; Koh & Liang, 2017). Moreover, the value of cosine similarity loss is bounded in $[-1, 1]$, simplifying hyper-parameter adjustment for reconstruction. An alternative type of loss could be the sum of element-wise distances, e.g., the Euclidean distance. And we show that cosine similarity loss outperforms Euclidean distance loss for data reconstruction in Appendix B.

Incorporating prior knowledge, represented by $\mathcal{X} \in \mathcal{F}$ abstractly, can further enhance reconstruction quality. For natural image classification problems, total variation (TV) is a widely-used term to describe the natural image manifold, which minimizes the variations between adjacent pixels to

promote image smoothness. Thus, we apply TV loss with $\alpha$ as the scaling factor to make the dummy images look more natural. Overall, the reconstruction loss is given as the following:

$$\mathcal{L}_{\mathrm{recon}}(\boldsymbol{x}, \hat{\boldsymbol{\theta}}_f; \boldsymbol{\theta}_0, \boldsymbol{\theta}_f) = -\frac{\langle \boldsymbol{\theta}_f - \boldsymbol{\theta}_0, \hat{\boldsymbol{\theta}}_f - \boldsymbol{\theta}_0 \rangle}{\|\boldsymbol{\theta}_f - \boldsymbol{\theta}_0\|\|\hat{\boldsymbol{\theta}}_f - \boldsymbol{\theta}_0\|} + \alpha \cdot \mathrm{TV}(\boldsymbol{x}). \tag{5}$$

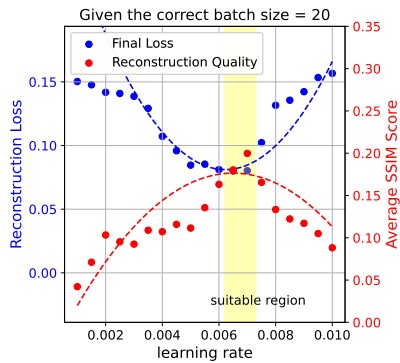

We test the designed loss in the reconstruction from a ResNet-18 trained on 60 images. The model is trained with a batch size of 20 and a learning rate of 7e-3. Fig. 2 plots the relationship between the final loss value (in blue) and reconstruction quality (in red) measured by average Structure Similarity Index Measure (SSIM) (Wang et al., 2004). In this figure, we fix the batch size but change the learning rates. One can observe that the loss and the quality have quite good consistency: lower loss values correspond to better reconstruction performances, showing the rationality of our loss design.

Figure 2: The relationship between final loss value and reconstruction quality.

**Gradient clipping.** We initialize dummy images with Gaussian noise. In the early stages of optimization, fully noisy images could cause gradient explosion. We apply gradient clipping in simulating dynamics to effectively prevent this issue. Additionally, even in the absence of gradient explosion, gradient clipping also simplifies image optimization. Since the training gradients of natural images generally have relatively small norms than noisy images, gradient clipping serves as a form of prior knowledge that guides the training process. Thus, this strategic use of gradient clipping enhances both the stability and efficiency of optimization for dataset reconstruction.

**Hyper-parameter settings.** The hyper-parameters $H$, including the batch-size $|\mathcal{B}|$ and the learning rate $\eta$, characterize the learning algorithm and thus partially influence the training process. Correct knowledge of $H$ facilitates reconstruction, while unknown $H$ make reconstruction more complex. In more practical cases, we need to design a strategy to set $H$ for better reconstruction.

Fig. 2 preliminarily shows that when the batch size $|\mathcal{B}|$ is fixed, one can do grid search for a good learning rate $\eta$ by comparing the final loss. And the yellow suitable region indicates that using learning rates within this area for simulating training can achieve good reconstruction. Here, we change the $|\mathcal{B}|$ and plot SSIM values for different $\eta$ in Fig. 3. Comparisons among the three figures show that the peak SSIM values across different $|\mathcal{B}|$ are similar. In other words, for different $|\mathcal{B}|$, as long as a suitable $\eta$ is paired, the quality is comparable to that obtained with the original $H$. Intuitively, when $|\mathcal{B}|$ decreases and the corresponding $\eta$ increases, similar final parameters can be obtained for the same dataset, and vice versa. Using grid search for $\eta$ suitable with a given $|\mathcal{B}|$, we can successfully reconstruct training data with unknown $H$.

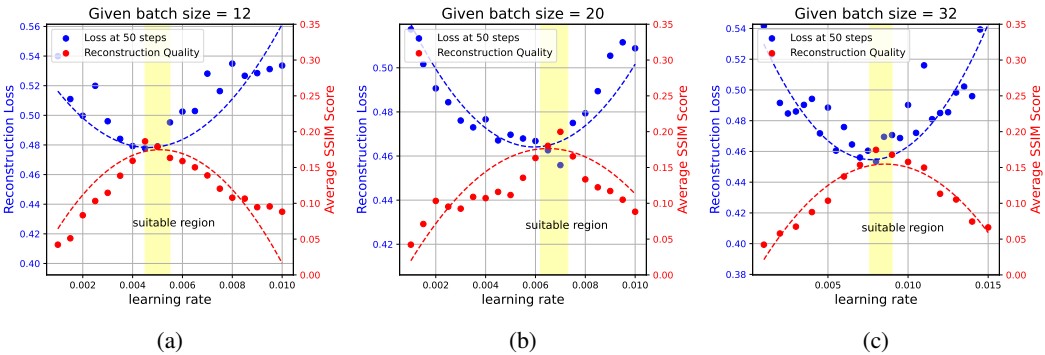

|  |  |  |
|:--:|:--:|:--:|
| (a) | (b) | (c) |

Figure 3: The relationship between the initial loss at 50 steps and the final reconstruction quality for different given batch sizes.

However, the reconstruction loss might take much time to converge. We experimentally explore how early decrease in loss within the first few dozen steps relates to the final reconstruction quality. As

shown in Fig. 3, lower value of loss at 50 steps (in blue) indicates better reconstruction performance (in red). In summary, we preset $|\mathcal{B}|$ and tune $\eta$ by grid search based on the first dozens of steps' loss. Thus, suitable $H$ for simulating the training dynamics can be identified effectively and efficiently.

# 4 EXPERIMENTS

## 4.1 EXPERIMENT SETUP

Our task is to reconstruct training data from parameters of DNNs. The experimental pipeline is as follows. First, we use the dataset $\mathcal{D}$ to train a model from the initial parameters $\boldsymbol{\theta}_0$ to the final parameters $\boldsymbol{\theta}_f$. Then, SimuDy is applied to reconstruct training data from the saved parameters $\boldsymbol{\theta}_0$ and $\boldsymbol{\theta}_f$. All training and reconstructing run on one RTX 4090 GPU. We provide the code in the supplementary materials.

**DNNs' Parameters.** We mainly consider the ResNet-18 model and will reconstruct data from its parameters. Following the setting of Loo et al. (2024), we reconstruct dataset from the initial and final model parameters. Consequently, We focus on the fine-tuning scheme, which is practical and popular in current applications. We train the model on a subset of CIFAR-10 (Krizhevsky et al., 2009) from pre-trained initial parameters $\boldsymbol{\theta}_0$ to the final parameters $\boldsymbol{\theta}_f$. The class distribution is balanced and the training algorithm is mini-batch gradient descent with shuffle on.

**Reconstructing.** Following the reconstructing protocol of previous works (Haim et al., 2022; Loo et al., 2024), the size of the original training set and the resolution of images are known for the reconstruction. To quantitatively evaluate the quality of the reconstructed images, we use SSIM (Wang et al., 2004) to measure the similarity between the reconstructed and original data, and pair them based on the SSIM scores. The average SSIM score is shown below the reconstructions, and the reconstructions are presented in descending order of SSIM scores.

## 4.2 RECONSTRUCTION PERFORMANCE

We first validate the effectiveness of our method on the reconstruction from MLPs, which are the primary focus of previous approaches (Haim et al., 2022; Buzaglo et al., 2024; Loo et al., 2024). The comparison is conducted to highlight the superiority of our approach. Since pre-training is meaningless for MLPs, we train MLPs, which comprise three fully-connected layers with dimensions d-1000-1000-1 (d is the dimension of the input), from scratch with SGD. As shown in Fig. 4, SimuDy gets an average 0.3374 SSIM score for reconstructions from the MLP trained on 100 images, outperforming Loo et al.'s method significantly.

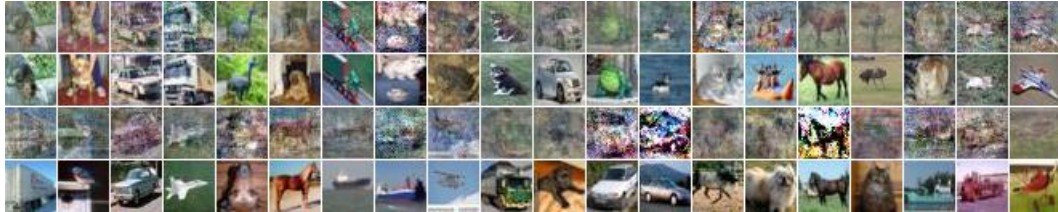

(a) Reconstructions from MLP trained on 100 images using Loo et al.'s, SSIM = 0.1384

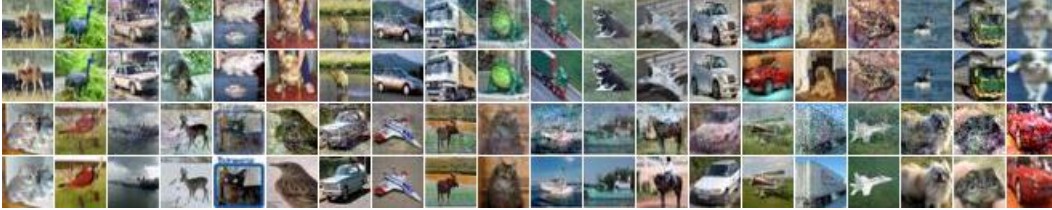

(b) Reconstructions from MLP trained on 100 images using SimuDy, SSIM = 0.3374

Figure 4: Top 40 images reconstructed from MLP trained on 100 images using Loo et al.'s and SimuDy (rows 1, 3) respectively, and corresponding nearest neighbors from the dataset (rows 2, 4).

The above shows that for MLPs, SimuDy has better reconstruction performance than those methods which are based on linear training dynamics. For more practical DNNs, the large number of parameters and complex structures, including convolutional layers and skip connections, make the training dynamics much more non-linear than MLPs. In this case, Loo et al.'s method only gets 0.0774 and SimuDy gets 0.1982 when reconstructing training data from ResNet trained on 50 images. The advantage on visual effect is also significant as shown in Fig. 5. And for a more comprehensive comparison against baselines, the reconstruction results of Buzaglo et al.'s method are presented in Appendix C.4.

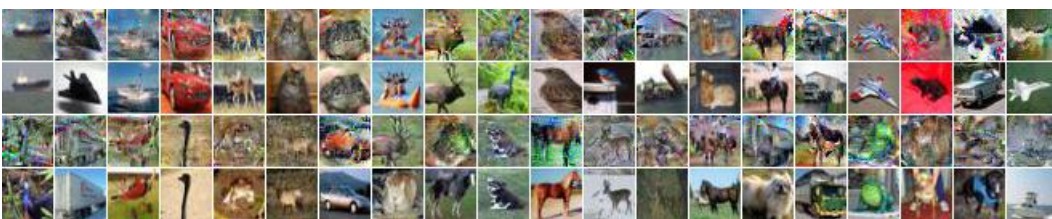

(a) Reconstructions from ResNet trained on 50 images using Loo et al.'s, SSIM = 0.0774

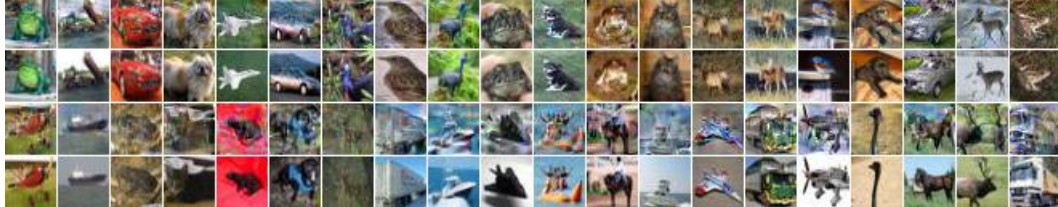

(b) Reconstructions from ResNet trained on 50 images using SimuDy, SSIM = 0.1982

Figure 5: Top 40 images reconstructed from ResNet trained on 50 images using Loo et al.'s and SimuDy (rows 1, 3) respectively, and corresponding nearest neighbors from the dataset (rows 2, 4).

Intuitively, as the training dataset size increases, the dataset reconstruction becomes more challenging, thus resulting in a decline in the reconstruction quality. For the reconstruction of 20 images shown in Fig. 1 and 50 images shown in Fig. 5, SimuDy can reconstruct nearly all data with high quality. And for 120 images shown in Fig. 6, SimuDy can also reconstruct about even 80 images. Moreover, as the size of dataset increases, the training dynamics become more non-linear, leading to a rapid decline in the reconstructing performance of Buzaglo et al.'s and Loo et al.'s method, which even fails when the dataset size reaches 120. The qualitative reconstruction results across variable-size datasets using different methods are presented in Appendix C.8.

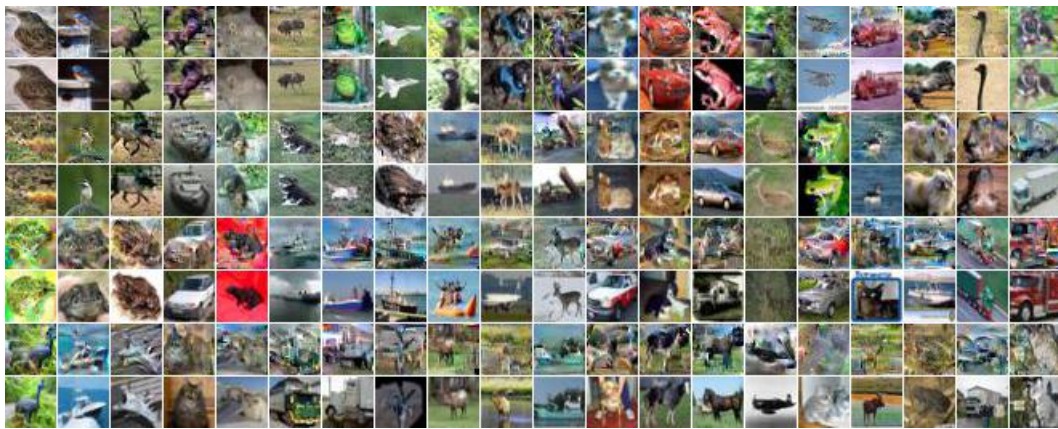

Figure 6: Top 80 images reconstructed from ResNet trained on 120 images using SimuDy (rows 1, 3, 5, 7), and their corresponding nearest-neighbors from the training dataset (rows 2, 4, 6, 8). The average SSIM score is 0.1196.

## 4.3 RECONSTRUCTIONS WITH UNKNOWN HYPER-PARAMETERS

In the above experiments, we assume that the batch size and the learning rate of the training process are known, which helps to describe the relationship between parameters and data. For more practical use, we design a strategy to identify a suitable learning rate for the given batch size which is specified in Section 3. Here, we report the reconstruction results from ResNet trained on 60 images with different training hyper-parameter settings in Fig. 7, from which one can observe that even the guess on batch-size is not accurate, the performance keeps good.

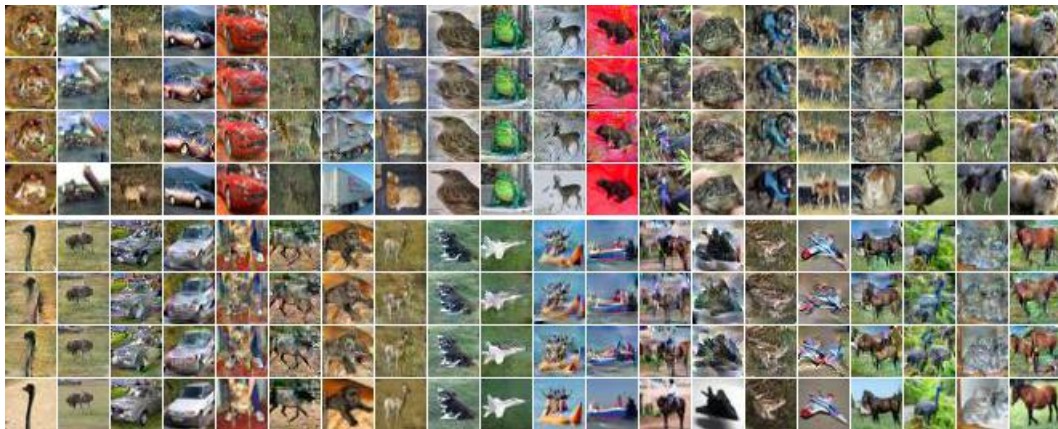

Figure 7: Reconstructions from ResNet trained on 60 images with batch size $|\mathcal{B}| = 20$. Different guesses of the batch size yield similar and good performance: Rows 1 and 5 are the images reconstructed by setting $|\mathcal{B}| = 20$; rows 2 and 6 are by setting $|\mathcal{B}| = 32$; rows 3 and 7 are by setting $|\mathcal{B}| = 12$, and rows 4 and 8 are the original training samples.

## 4.4 INITIALIZATION OF DUMMY IMAGES

Different initializations of dummy images would influence the reconstruction quality. In the above experiments, we use random Gaussian noise as the initialization. Another seemingly reasonable choice is to use other natural images. Fig. 8 illustrates the reconstruction process from the two different initialization. One can see that although the main object can reconstructed from natural image initialization, the background seems be seriously influenced by the initialization. For example, the background of the reconstructed cat (row 2) is more brighter since the initial image is a white dog. The reason may be the fact that a trained classification model mainly focuses on the main object, and changes in the background minimally affect the gradients, thus the parameter changes carry less information of backgrounds than that of main body.

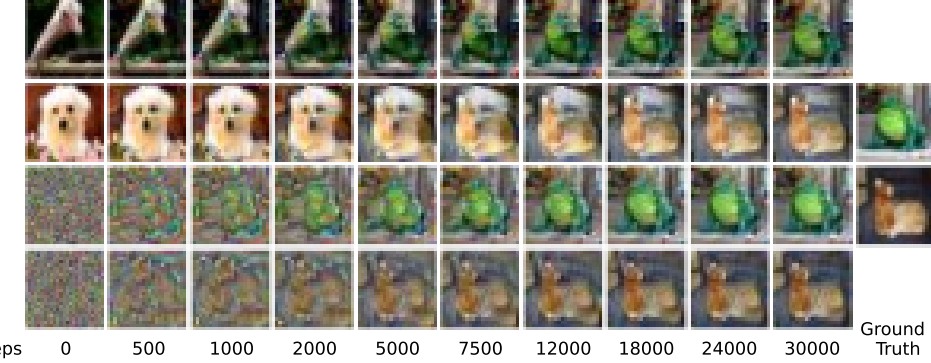

Steps   0   500   1000   2000   5000   7500   12000   18000   24000   30000   Ground Truth

Figure 8: The reconstruction process of different initializations. The last column shows the ground truth, while in the other columns the top 2 images are intervals initialized from natural images, and the bottom 2 images from random noise.

## 4.5 RECONSTRUCTIONS OF DUMMY DATASETS OF DIFFERENT SIZES

In previous works (Haim et al., 2022; Buzaglo et al., 2024; Loo et al., 2024), the size of training dataset is necessitated for effective dataset reconstruction. In this section, We evaluate the effectiveness of our method without known original dataset size.

Given parameters of the ResNet trained on 40 images, we construct dummy datasets of different sizes for reconstructions. When the size of dummy dataset is 20, smaller than that of the original datset, the cosine similarity loss of SimuDy fluctuates around 0.2 while good reconstructions always have losses below 0.1. This suggests that no dataset with merely 20 samples can guide the model to traverse from $\boldsymbol{\theta}_0$ to $\boldsymbol{\theta}_f$. The reconstructions are shown in the left of Fig. 9. Although these images appear unconventional, they still contain clues of the original data. As shown in the right part of Fig. 9, the reconstructed image seems like the fusion of two frog images from the original dataset.

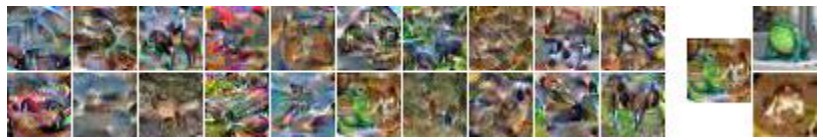

Figure 9: Reconstructions from ResNet trained on 40 images using a dummy dataset of size 20.

With a larger dummy dataset of size 60, the loss value can be refined to about 0.08, indicating a well match of training dynamics. The images from the original dataset are paired with their nearest neighbors from the dummy dataset based on the SSIM scores, leaving remaining images in the dummy dataset unmatched. As shown in Fig. 10(a), we successfully reconstruct the original dataset of size 40 using a dummy dataset of size 60. And the unmatched images are shown in Fig. 10(b), appearing also unconventional but different from those in Fig. 9, as there are few clues of original data in unmatched images.

The contribution of each data point from the dummy dataset to the parameter changes is calculated by the norm of overall gradients throughout the training dynamics. We observe that the matched data's average norm of total gradients is 1.5458, ranging from 1.1204 to 1.9478. In contrast, for unmatched data, the average norm is only 0.5466, with extremes of 0.2172 to 0.8438. These findings validate our approach's ability to reconstruct dataset with a larger size setting and transform extra dummy images from random noise to insignificant images which have relatively small gradients during training.

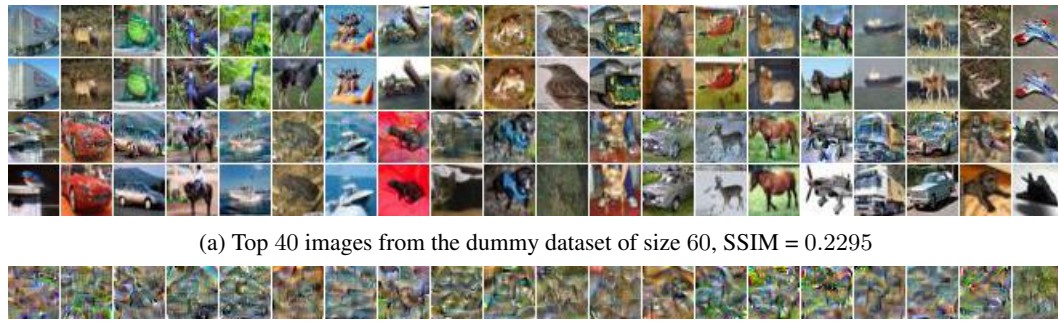

(a) Top 40 images from the dummy dataset of size 60, SSIM = 0.2295

(b) Bottom 20 images from the dummy dataset of size 60

Figure 10: Reconstructions from ResNet trained on 40 images using a dummy dataset of size 60.

With a smaller dataset size, our approach tries to amalgamate multiple images' information into a single image, which compromises the quality of recovery. However, when handling unknown dataset size, we can increase the size of dummy dataset as SimuDy can reconstruct original dataset with high quality while optimizing extra dummy images into insignificant images.

## 5 Conclusion, Limitation, and Future Work

In this work, we propose SimuDy to reconstruct training data from the parameters of DNNs. This reconstruction task may open up possibilities to the study on the memorization of models which is quite challenging. Previous works are restricted to simple NNs with linear training dynamics, while our method can be used for ResNet, towards more practical and meaningful reconstructing. The core progress is that SimuDy simulates the training dynamics which are crucial for reconstructing training data, since parameters are adjusted iteratively by data during training. Successful reconstructions illustrate the importance of characterizing training dynamics for extracting data from parameters, which provides insights for memorization of DNNs and helps better understand deep learning.

Our proposed method, SimuDy, successfully reconstructs training data from parameters of DNNs, though it does not completely solve dataset reconstruction. We acknowledge the limitations of our approach and encourage further research.

One primary limitation is that our method's performance declines as the dataset size increases, due to the increased uncertainty in the optimization problem of decoupling gradients of more data. This highlights the need for more effective solutions to more complex dataset reconstruction optimization challenges. Additionally, the simulation of training dynamics necessitates the preservation of the entire model training computation graph. Thus our method is restricted to the GPU memory when dealing with large datasets. Future efforts should focus on characterizing training dynamics more efficiently, thus minimizing computational resource demands, such as by leveraging the low-dimensional nature of the model training process. Last but not least, although our method could reconstruct about only 100 images from parameters, this is sufficient to show that indeed there is memorization in DNNs, providing a promising tool for investigating deep learning memory. We hope our work will illuminate deep learning interpretability and stimulate further exploration into the relationship between memorization and generalization of DNNs on larger datasets.

### Acknowledgments

The authors would like to thank the anonymous reviewers for their insightful comments.

The research leading to these results has received funding from National Natural Science Foundation of China (62376155), Shanghai Municipal Science and Technology Research Program Major Project (2021SHZDZX0102).

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

# APPENDIX

## A  TRAINING DYNAMICS LINEARITY

Table 1: Comparison of training dynamics linearity metrics ($\mathcal{M}_{\text{lin}}$) for 2-layer MLP with varying widths and ResNet-18, all trained on 100 CIFAR-10 images with a fixed learning rate of 0.01 and batch size 100. A higher $\mathcal{M}_{\text{lin}}$ indicates a more linear training dynamics.

| Model | Width | $\mathcal{M}_{\text{lin}}(\uparrow)$ |
|---|---|---|
| MLP | 200 | 0.9134 |
| | 500 | 0.9165 |
| | 1000 | 0.9306 |
| | 2000 | 0.9396 |
| | 4000 | 0.9535 |
| ResNet-18 | / | 0.5988 |

To measure the linearity of training dynamics, we calculate the cosine similarity between gradient pairs for $m$ samples across different training epochs in dataset $\mathcal{D}$:

$$\mathcal{M}_{\text{lin}} = \frac{2}{m} \frac{1}{T(T-1)} \sum_{i=1}^{m} \sum_{1 \leq t_1 \neq t_2 \leq T} \frac{\langle g_{i,t_1}, g_{i,t_2} \rangle}{\|g_{i,t_1}\| \|g_{i,t_2}\|}, \tag{6}$$

where $g_{i,t}$ represents the gradient of the $i$-th sample at $t$-th epoch. And $\mathcal{M}_{\text{lin}}$ lies within $[-1, 1]$. A value of 1 indicates that the gradients across different epochs are perfectly aligned, which is characteristic of a completely linear network.

As illustrated in Tab. 1, the linearity of MLP increases as network width expands, which aligns with the theoretical predictions of neural tangent kernel (NTK) (Jacot et al., 2018) that infinitely wide MLP tends to a linear network. In contrast, ResNet-18 exhibits significantly lower linearity in training, due to its deeper and more complex structure.

## B  LOSS CHOICE

We use Euclidean distance loss to replace cosine similarity loss for reconstructions of 20 images. Due to the large variations in Euclidean distance loss during optimization, it requires careful learning rate adjustments, and the final reconstruction quality is low compared to cosine similarity loss. The qualitative results of Euclidean distance loss is shown in Fig. 11 with an average SSIM score of 0.2437, while results of cosine similarity loss shown in Fig. 1 get 0.3140 SSIM score.

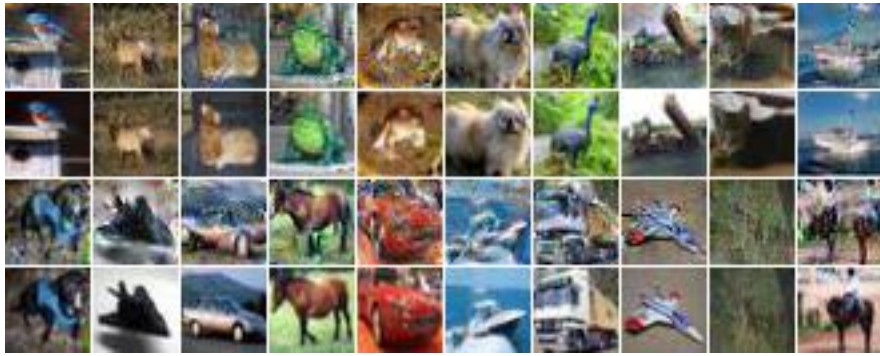

Figure 11: Reconstructed training samples from a multi-class ResNet (rows 1, 3) using Euclidean distance loss, and the corresponding nearest-neighbors from the training dataset (rows 2, 4).

## C    ADDITIONAL EXPERIMENTS

### C.1    RECONSTRUCTIONS FOR LARGER IMAGES

While previous experiments are conducted on CIFAR-10 with a resolution of $32 \times 32$, SimuDy can be scaled to higher resolutions. As shown in Fig. 12, the reconstruction results on the $64 \times 64$ Tiny ImageNet demonstrate the potential of our approach at higher resolutions.

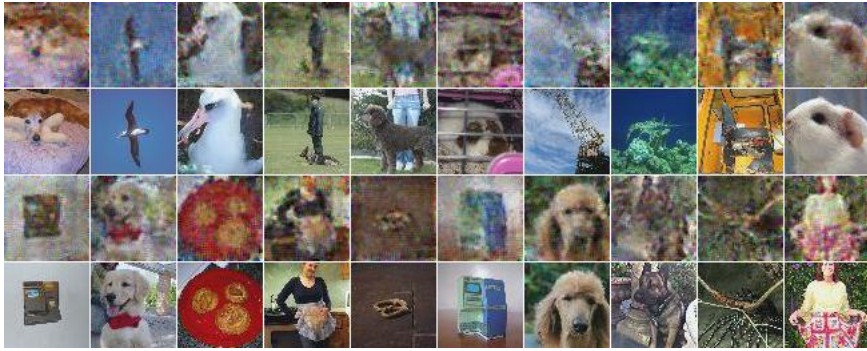

Figure 12: Reconstructions from ResNet-18 trained on Tiny ImageNet (rows 1, 3) using SimuDy, and the corresponding nearest-neighbors from the training dataset (rows 2, 4).

### C.2    RECONSTRUCTIONS ON OTHER DATASET AND ARCHITECTURE

To further demonstrate that SimuDy maintains its effectiveness across various datasets and diverse network architectures, we extend our experiments to SVHN with ResNet-18 and CIFAR-10 with ResNet-50. The reconstruction results are presented in Fig. 13 and Fig. 14, respectively. Additionally, we conduct experiments on ImageNet with ViT and an NLP task, as shown in Appendix C.6 and Appendix C.7. In all these experiments, SimuDy consistently achieves strong performance, highlighting its robustness across different data distribution and network architectures.

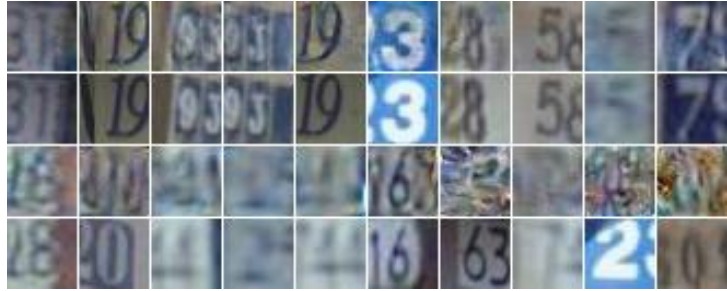

Figure 13: Reconstructions from ResNet-18 using SimuDy on SVHN, SSIM = 0.2083.

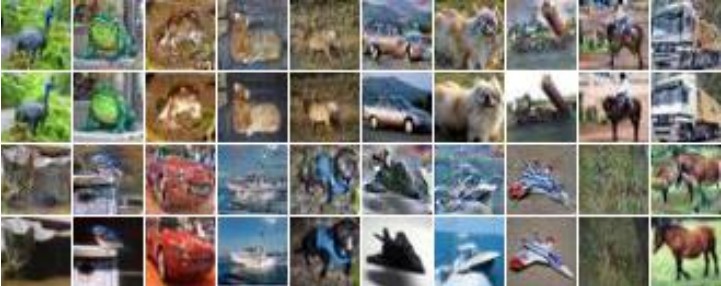

Figure 14: Reconstructions from ResNet-50 using SimuDy on CIFAR-10, SSIM = 0.2125.

## C.3 Affection of TV Loss

We test how the coefficient of TV loss affects the quality by discretely selecting the coefficient on reconstructing 20 images from a trained ResNet-18. We set the coefficients to 0, 5e-4, 2e-3, and 5e-2, respectively. When the coefficient is 2e-3, the average SSIM score can reach $0.3140$, with qualitative results shown in Fig. 1. And other quantitative and qualitative reconstruction results are shown in Fig. 15, Fig. 16 and Fig. 17.

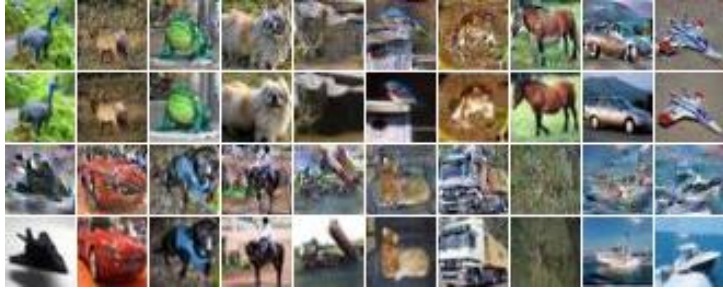

Figure 15: Reconstructions from ResNet with TV loss coefficient of 0, SSIM = 0.1835.

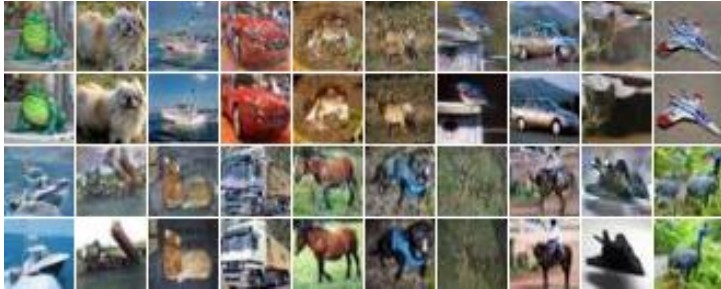

Figure 16: Reconstructions from ResNet with TV loss coefficient of 5e-4, SSIM = 0.2555.

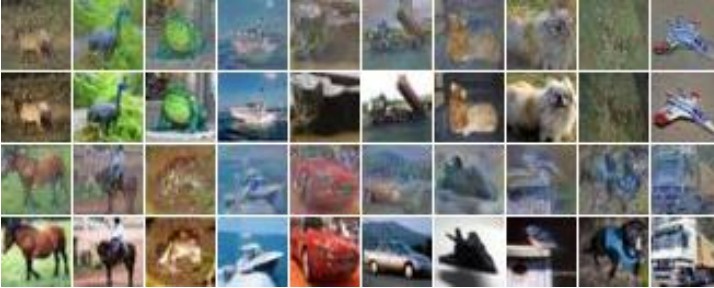

Figure 17: Reconstructions from ResNet with TV loss coefficient of 5e-2, SSIM = 0.2255.

The coefficient of TV loss has affects on the final reconstruction quality. Incorporating TV loss will help achieve better compared to not using it. Also, an appropriate coefficient is important, as both overly large and small values can degrade the reconstruction quality. To be noticed, an excessively large coefficient can cause reconstructed images to become blurry, and may even result in color shifts compared to original training images, as shown in Fig. 17.

## C.4 Comparison with Buzaglo et al.'s Method

To achieve a comprehensive comparison against baseline methods, we test the performance of Buzaglo et al.'s method on reconstructing images from parameters of models. We first present the quantitative and qualitative reconstructions from MLPs trained on 100 images in Fig. 18, serving as a comparative counterpart to Fig. 4.

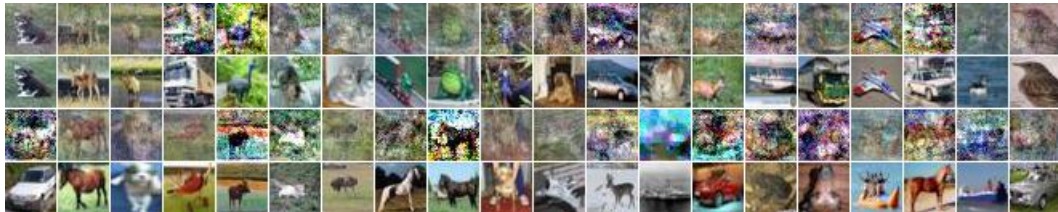

Figure 18: Top 40 images reconstructed from MLP trained on 100 images using Buzaglo et al.'s, SSIM = 0.1426.

As illustrated in Fig. 18, while Buzaglo et al.'s method demonstrates the capability to partially reconstruct training dataset from the parameters of MLPs, it falls short of our SimuDy in terms of both reconstructing quantity and quality. Subsequently, we demonstrate the performance of Buzaglo et al.'s method when applied to deep neural networks. The reconstruction results from ResNet-18 trained on 20 images are presented in Fig. 19, enabling direct comparison with Fig. 1, while reconstructions from ResNet-18 trained on 50 images are shown in Fig. 20, corresponding to Fig. 5.

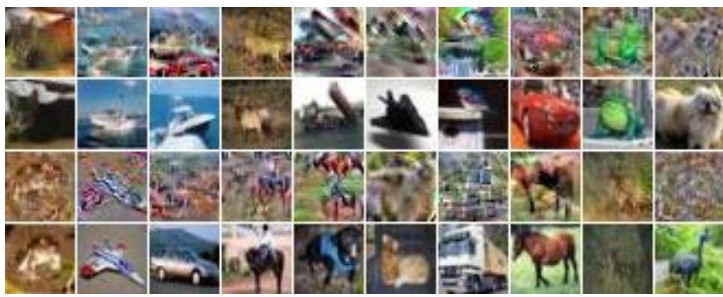

Figure 19: Reconstructions from ResNet using Buzaglo et al.'s method, SSIM = 0.0476.

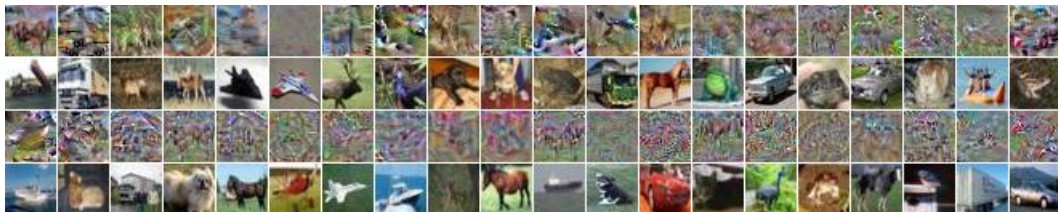

Figure 20: Top 40 images reconstructed from ResNet trained on 50 images using Buzaglo et al.'s, SSIM = 0.0297

When applied to ResNet-18 with non-linear training dynamics, Buzaglo et al. (2024) use only gradients of final models to characterize the whole dynamics, which is inadequate and degrades reconstruction quality. Also, their method requires very small initial parameters or the use of weight decay in training, and the absence of these factors could further degrade the performance.

## C.5 RECONSTRUCTIONS OF BINARY CLASSIFICATION SETTING

In this section, we manage to reconstruct training dataset from models trained for the binary classification task. Following the pioneering work (Haim et al., 2022), we use CIFAR-10 dataset and set the labels to vehicles vs. animals. Also, we make sure that the class distribution in the training and test sets is balanced. For a comprehensive comparison with baselines, we use Haim et al.'s method, Loo et al.'s method and SimuDy to reconstruct dataset from a MLP trained on 20 images, with quantitative and qualitative results shown in Fig. 21a, Fig. 21b and Fig. 21c, respectively. Both quantitative and qualitative results show that SimuDy can well applied to binary classification setting.

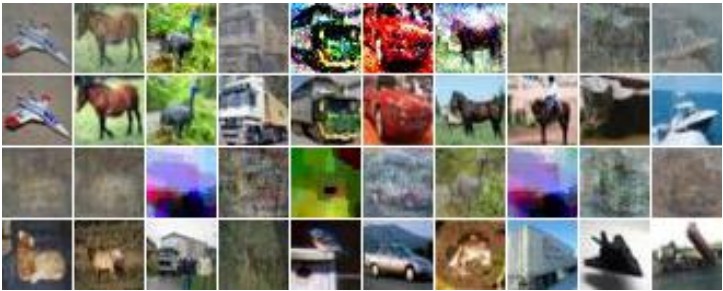

(a) Reconstructions in binary classification setting using Haim et al.'s, SSIM = 0.2276

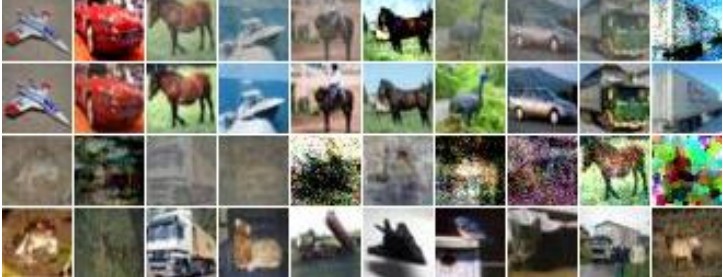

(b) Reconstructions in binary classification setting using Loo et al.'s, SSIM = 0.3537

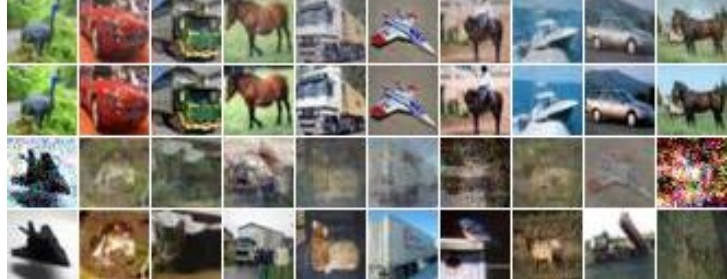

(c) Reconstructions in binary classification setting using SimuDy, SSIM = 0.5378

Figure 21: Reconstructions from a trained binary MLP trained on 20 images using different methods.

## C.6 RECONSTRUCTIONS FOR VISION TRANSFORMERS

To evaluate the generalizability of SimuDy, we conduct experiments on Vision Transformers (ViTs) (Dosovitskiy, 2021), which are larger and have different architectures compared to ResNet. For the original training dataset, we choose ImageNet with a resolution of $224 \times 224$. Inspired by APRIL (Lu et al., 2022), we noticed that the input $z$ of the self-attention module is important for reconstruction but is invisible from attacker's side. However, for a transformer-based model with learnable position embedding $E_{pos}$, the derivative of loss w.r.t. $E_{pos}$ can be given by:

$$\frac{\partial l}{\partial E_{pos}} = \frac{\partial l}{\partial z}$$

Based on this, we update the reconstruction loss for ViTs as following:

$$\mathcal{L}_{\mathrm{ViT}}(\boldsymbol{x}, \hat{\boldsymbol{\theta}}_f; \boldsymbol{\theta}_0, \boldsymbol{\theta}_f) = \mathcal{L}_{\mathrm{recon}}(\boldsymbol{x}, \hat{\boldsymbol{\theta}}_f; \boldsymbol{\theta}_0, \boldsymbol{\theta}_f) - \beta \cdot \frac{\langle \boldsymbol{\theta}_{f,E_{pos}} - \boldsymbol{\theta}_{0,E_{pos}}, \hat{\boldsymbol{\theta}}_{f,E_{pos}} - \boldsymbol{\theta}_{0,E_{pos}} \rangle}{\|\boldsymbol{\theta}_{f,E_{pos}} - \boldsymbol{\theta}_{0,E_{pos}}\|\|\hat{\boldsymbol{\theta}}_{f,E_{pos}} - \boldsymbol{\theta}_{0,E_{pos}}\|},$$

where $\boldsymbol{\theta}_{0,E_{pos}}$ represents the initial parameters of the position embedding module, and $\boldsymbol{\theta}_{f,E_{pos}}$ refers to the final parameters of the position embedding module after training. The second term of the loss function effectively helps in ensuring the correct positioning of the patches.

Both training and reconstructing are conducted on one RTX 4090 GPU. We successfully reconstruct 10 ImageNet images from the parameters of a trained ViT, with the qualitative reconstruction results

shown in Fig. 22. The successful reconstructions validate the scalability of SimuDy to larger images and transformer architectures.

Figure 22: Reconstructions of 10 ImageNet images from a trained ViT.

### C.7 RECONSTRUCTIONS IN NLP REGIME

In this section, we extend dataset reconstruction from image classification task to the natural language processing (NLP) regime. We choose TinyBERT (Jiao et al., 2020) as model and CoLA (Warstadt et al., 2018) as dataset, with sentence classification as the training task. Both training and reconstructing run on one RTX 4090 GPU. Following the state-of-the-art gradient inversion attack method in NLP (Balunovic et al., 2022), which recovers a size-of-4 batch of data from the gradient, we apply SimuDy to simulate the training dynamics and successfully reconstruct sentence data from parameters of trained TinyBERT parameters. The qualitative reconstructions are shown in Tab. 2.

### C.8 QUALITATIVE RESULTS FOR ORIGINAL DATASETS OF VARIOUS SIZES

In this section, we vary the size of original training dataset and show the qualitative and quantitative results placed side by side for each baseline method and our proposed SimuDy, as illustrated from Fig. 23 to Fig. 40. Also, we report GPU memory usage and reconstruction time of SimuDy on one RTX 4090 GPU for CIFAR-10 datasets with different sizes in Tab. 3.

| Original | Initial | Reconstructed |
|---|---|---|
| [CLS] who do you think that will question seamus first? [SEP] | [CLS]quistanger fixingimeter cpc forbidden nehru tread terminology [SEP] | [CLS] who do you think will do first question seamus? [SEP] |
| [CLS] the boy ran. [SEP] | [CLS] [PAD] [PAD] [SEP] [PAD] foo nightmares [PAD] [PAD] 102 drawer [PAD] | [CLS] the boy ran. [SEP] |
| [CLS] i wonder who bill saw and liked mary. [SEP] | [CLS] essays carltonomy crestedhiskhandnac [SEP] vita gail [PAD] | [CLS] i saw mary wonder who liked bill anything. [SEP] |
| [CLS] harriet alternated folk songs and pop songs together. [SEP] | [CLS] donetsk dominance grossedlok ass somerset registrar rochdale ins cher [SEP] | [CLS] harriet alternated alternate folk and pop songs together. [SEP] |
| [CLS] they have no old. [SEP] | [CLS] [PAD]oric [PAD] revealing [PAD] destroyer [PAD] louisiana [PAD] [SEP] [PAD] | [CLS] they have no old. [SEP] |
| [CLS] which goddess helped us? [SEP] | [CLS] [PAD] [PAD] pathways [PAD] [PAD] macon [PAD]emy theresa [PAD] [PAD] [PAD]rod [PAD] [SEP] [PAD] [PAD] | [CLS] goddess which helped us? [SEP] |
| [CLS] who has seen my snorkel? [SEP] | [CLS] [PAD] [PAD] 2011 [PAD] [SEP] articulated caslink [PAD] [PAD] implementskins [PAD] via [PAD] [PAD] | [CLS] who seenkel mynorkel? [SEP] |
| [CLS] that the king or queen be present is a requirement on all royal weddings. [SEP] | [CLS] sorbonne citationsimeter tsar citizens state accumulate jared sorbonne racecourse portraying perkins differ deco [SEP] | [CLS] the queen or a requirement on all that present is the royal weddings. [SEP] |
| [CLS] i saw these dancers and those musicians smoking something. [SEP] | [CLS] feminism diagnosticnell andover delicately strikingcade directorxing vfl [SEP] | [CLS] i saw these dancers and those musicians smoking something. [SEP] |
| [CLS] andy promised that we would go. [SEP] | [CLS] [PAD] laurent unconstitutionalhony [PAD] demi duncan kristin [SEP] [PAD] | [CLS] we promised that andy would go. [SEP] |
| [CLS] the cat tries to be out of the bag. [SEP] | [CLS] [PAD] unless denoted leiden harm answering [SEP]ides [PAD]ible hoffman [PAD] [PAD] plata [PAD] | [CLS] the be the cat tries out of the bag. [SEP] |
| [CLS] we talked about that he had worked at the white house. [SEP] | [CLS] siena tomatoes shakeettes sachsaseacy mosesasco flinders [SEP] | [CLS] we talked about that he had worked at the white house. [SEP] |
| [CLS] jessica sprayed paint under the table. [SEP] | [CLS] [PAD] reborn [PAD] zhao [SEP] leukemia braking vamp [PAD] matchkled [PAD]olar [PAD] | [CLS] jessica spray under sprayed table paint. [SEP] |
| [CLS] mary noticed john's excessive appreciation of himself. [SEP] | [CLS] onward atoms macro undo sweets rounding sparselyde cornerbackomorphic [SEP] | [CLS] and noticed mary. s excessive appreciation of john himself [SEP] |
| [CLS] kim alienates cats and beat his dog. [SEP] | [CLS] qui barred posed reverend wasn disputetilityphysics rabbis [SEP] [PAD] | [CLS] kim alienates cats and beat his dog. [SEP] |

Table 2: Reconstructed samples in NLP regime. The first column contains the original sentences of the training dataset, the second column contains the initializations of dummy sentences, and the third column presents the corresponding reconstructed sentences.

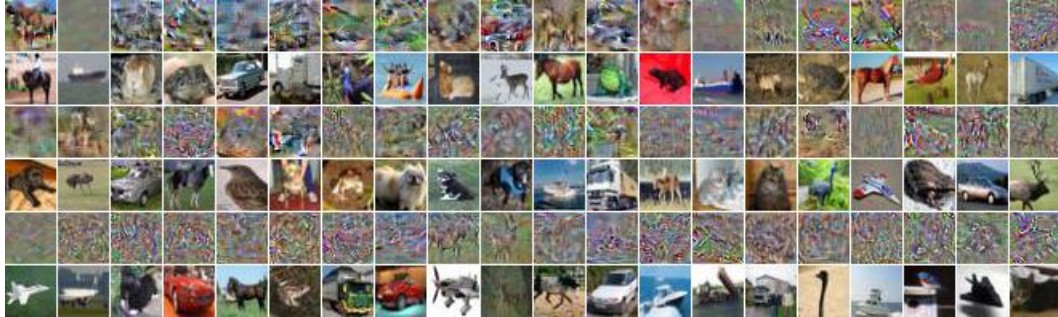

Figure 23: Reconstructions from ResNet trained on 60 images using Buzaglo et al.'s method, SSIM = 0.0285.

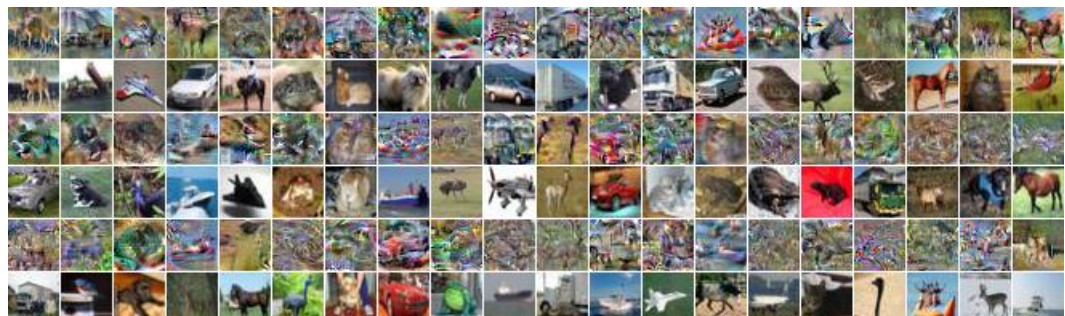

Figure 24: Reconstructions from ResNet trained on 60 images using Loo et al.'s method, SSIM = 0.0423.

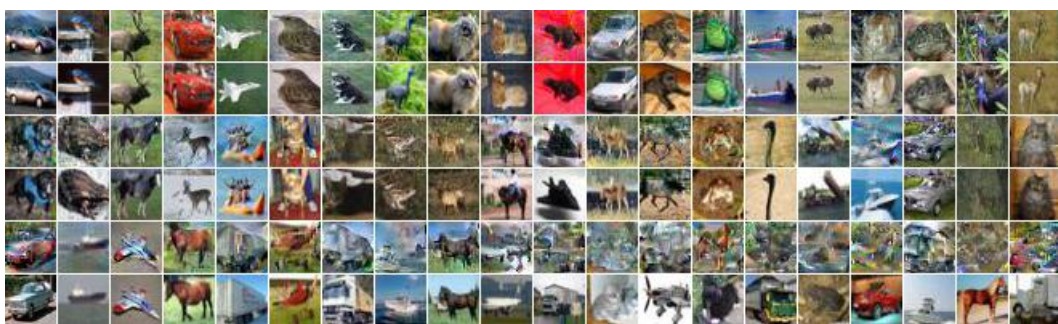

Figure 25: Reconstructions from ResNet trained on 60 images using SimuDy, SSIM = 0.1998.

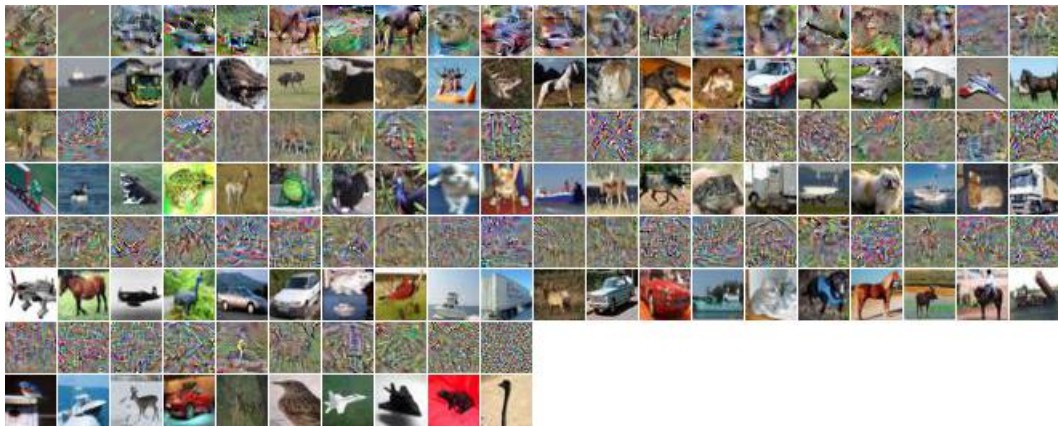

Figure 26: Reconstructions from ResNet trained on 70 images using Buzaglo et al.'s method, SSIM = 0.0286.

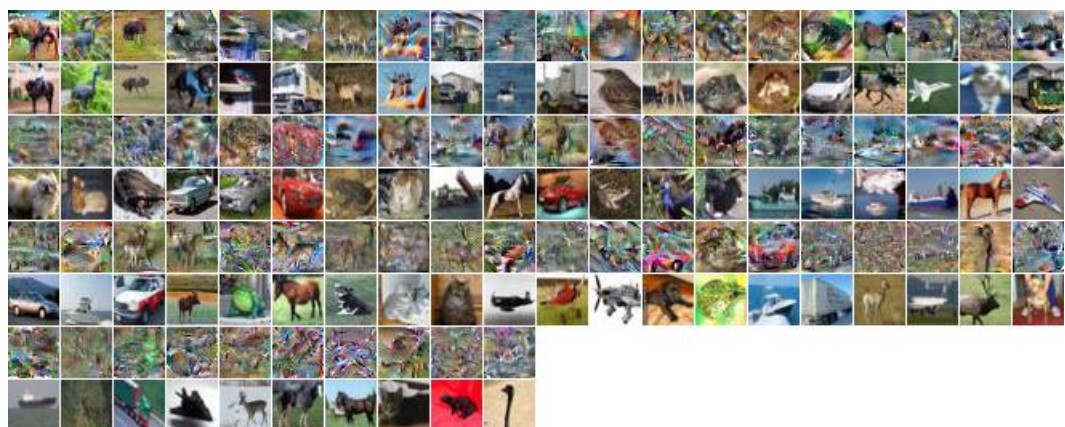

Figure 27: Reconstructions from ResNet trained on 70 images using Loo et al.'s method, SSIM = 0.0424.

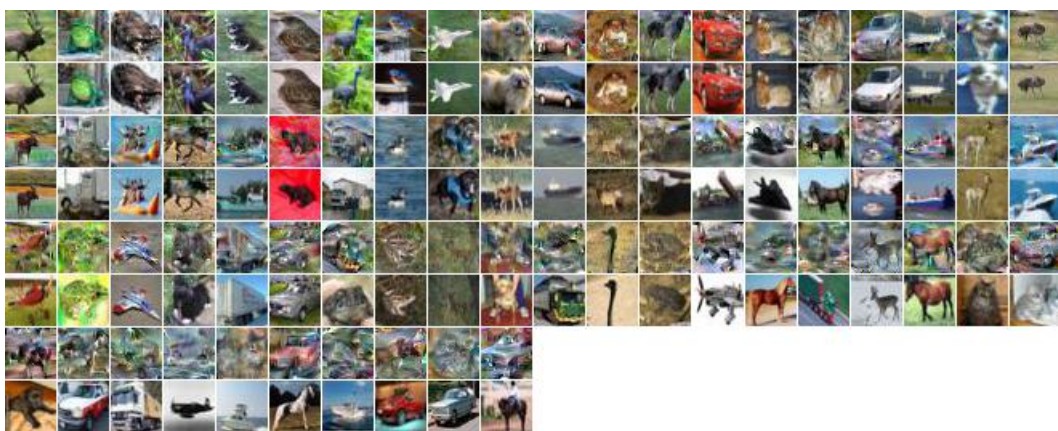

Figure 28: Reconstructions from ResNet trained on 70 images using SimuDy, SSIM = 0.1800.

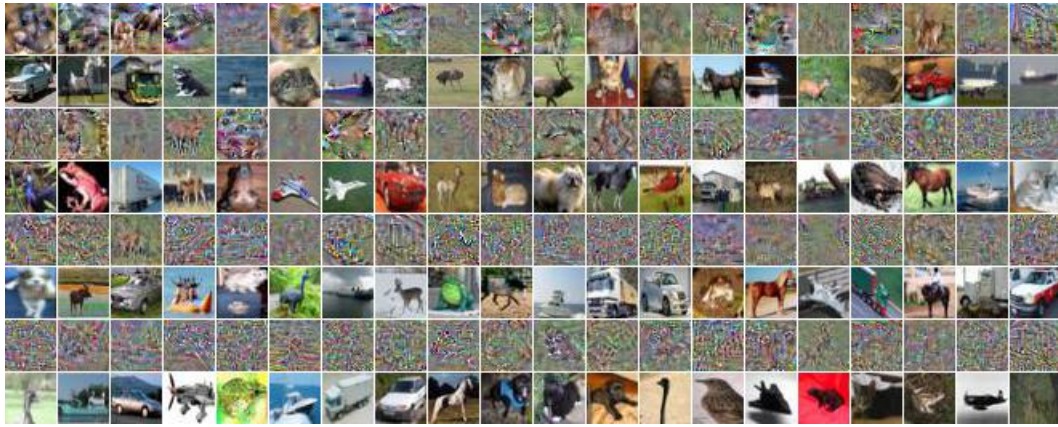

Figure 29: Reconstructions from ResNet trained on 80 images using Buzaglo et al.'s method, SSIM = 0.0253.

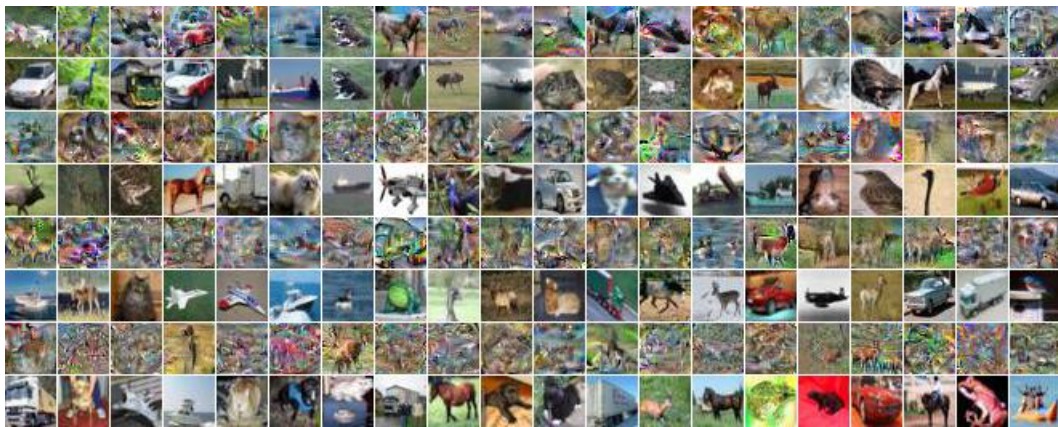

Figure 30: Reconstructions from ResNet trained on 80 images using Loo et al.'s method, SSIM = 0.0462.

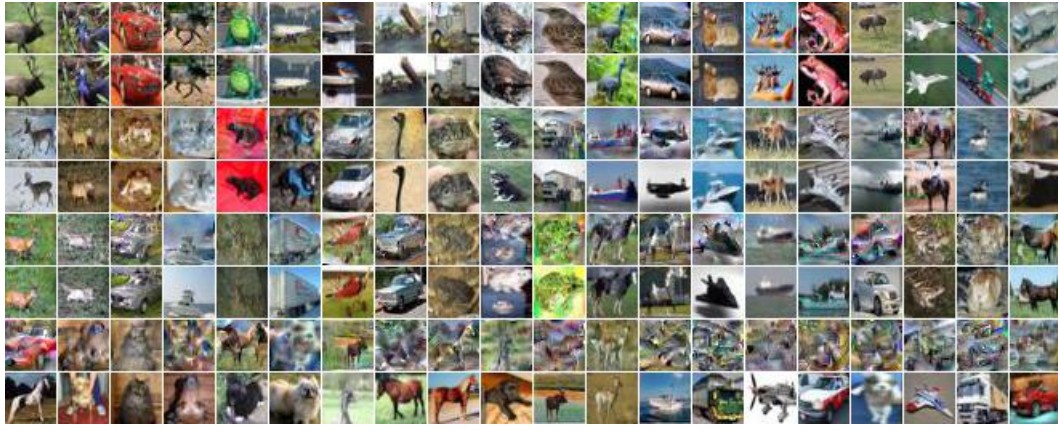

Figure 31: Reconstructions from ResNet trained on 80 images using SimuDy, SSIM = 0.2165.

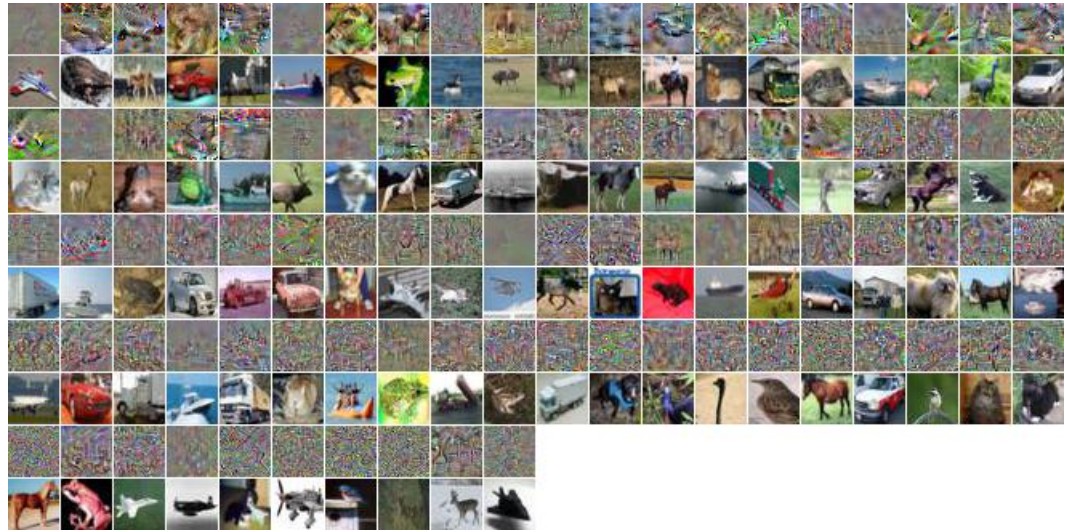

Figure 32: Reconstructions from ResNet trained on 90 images using Buzaglo et al.'s method, SSIM = 0.0249.

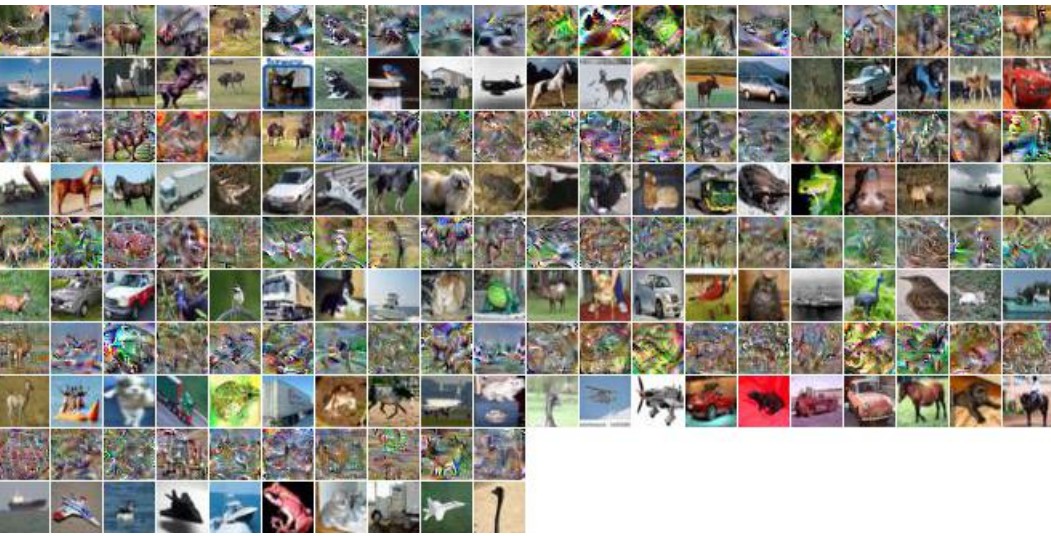

Figure 33: Reconstructions from ResNet trained on 90 images using Loo et al.'s method, SSIM = 0.0442.

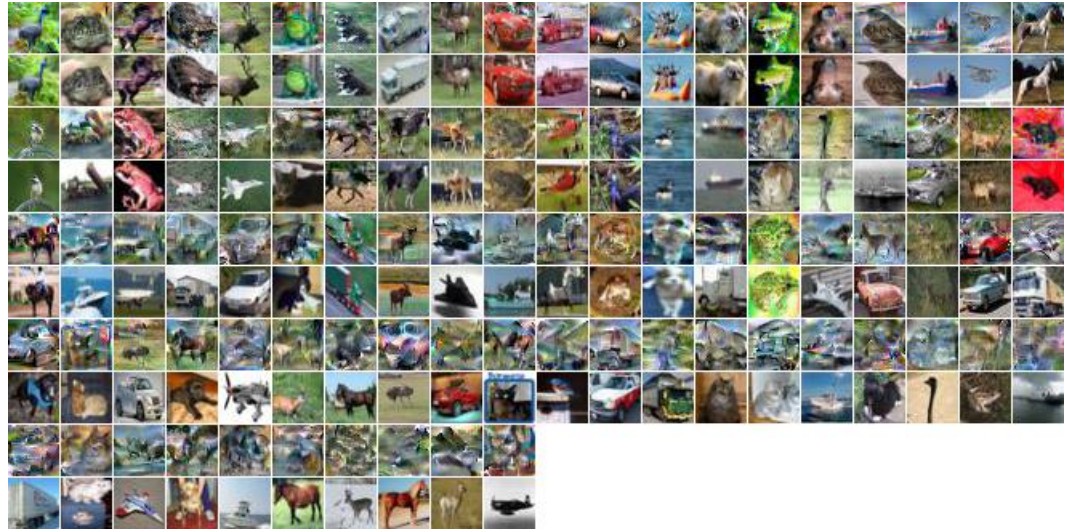

Figure 34: Reconstructions from ResNet trained on 90 images using SimuDy, SSIM = 0.1289.

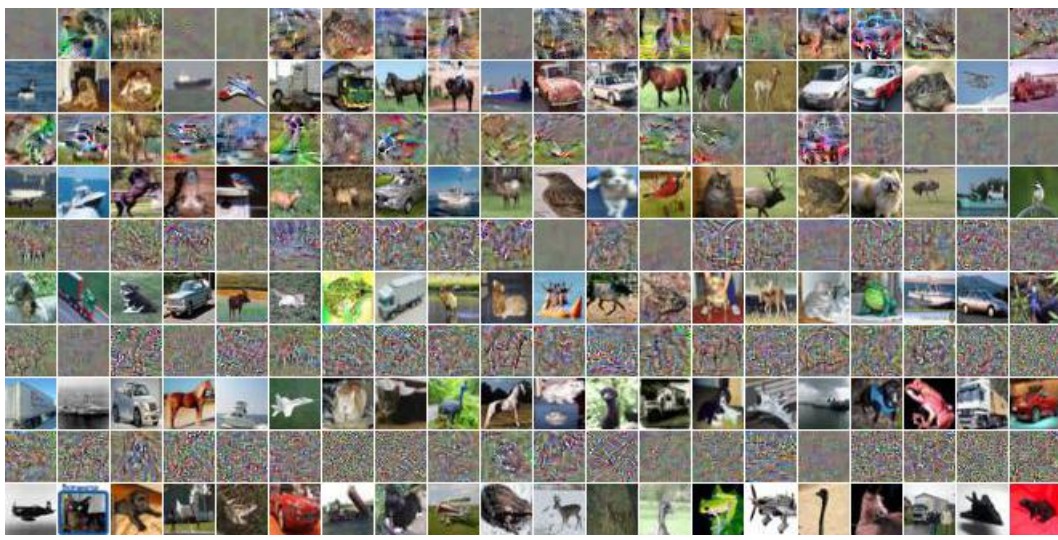

Figure 35: Reconstructions from ResNet trained on 100 images using Buzaglo et al.'s method, SSIM = 0.0267.

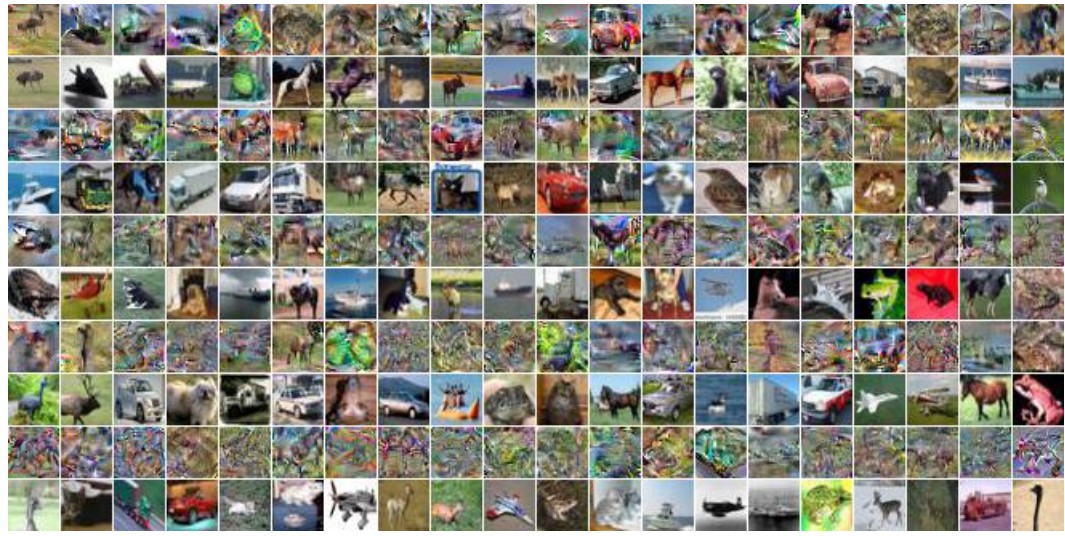

Figure 36: Reconstructions from ResNet trained on 100 images using Loo et al.'s method, SSIM = 0.0476.

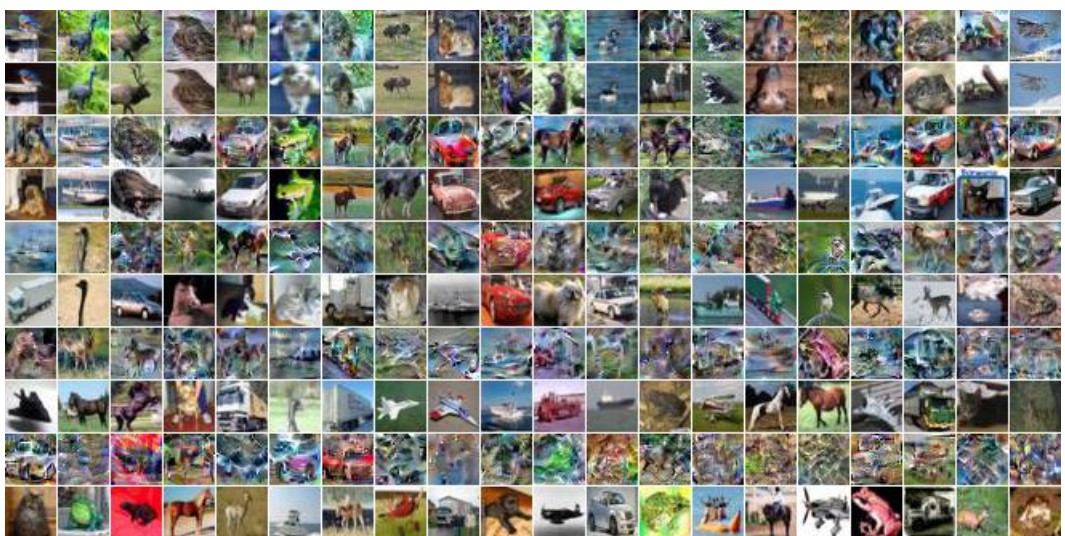

Figure 37: Reconstructions from ResNet trained on 100 images using SimuDy, SSIM = 0.0948.

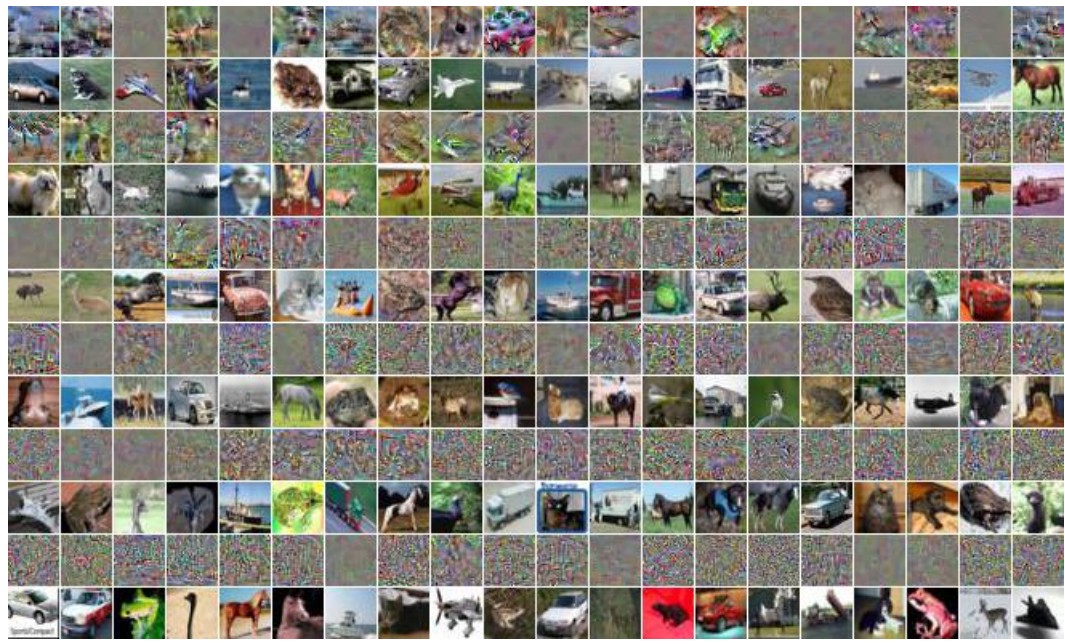

Figure 38: Reconstructions from ResNet trained on 120 images using Buzaglo et al.'s method, SSIM = 0.0239.

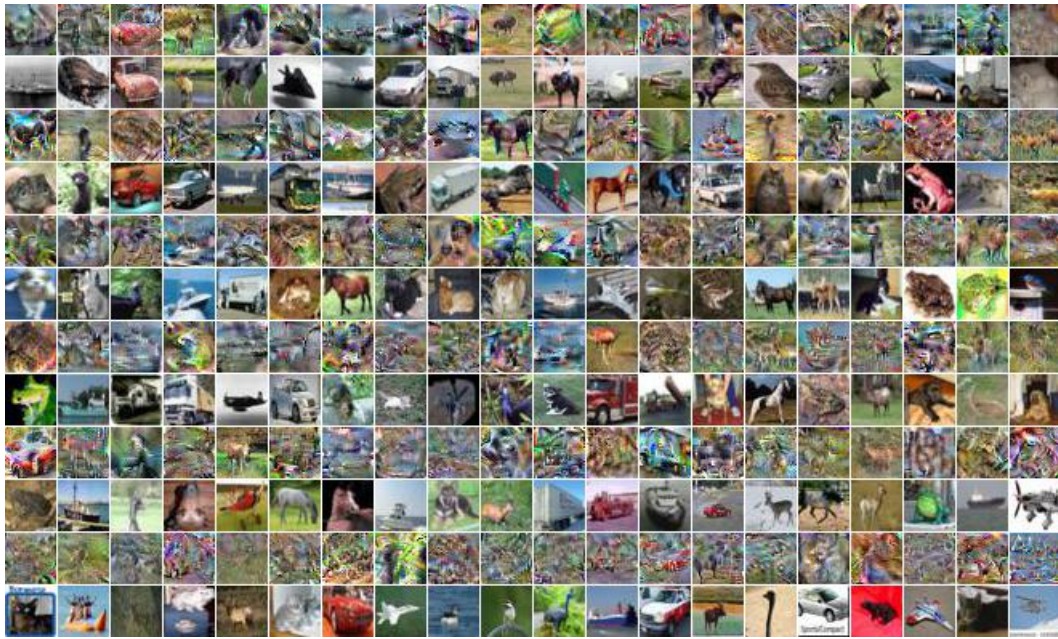

Figure 39: Reconstructions from ResNet trained on 120 images using Loo et al.'s method, SSIM = 0.0482.

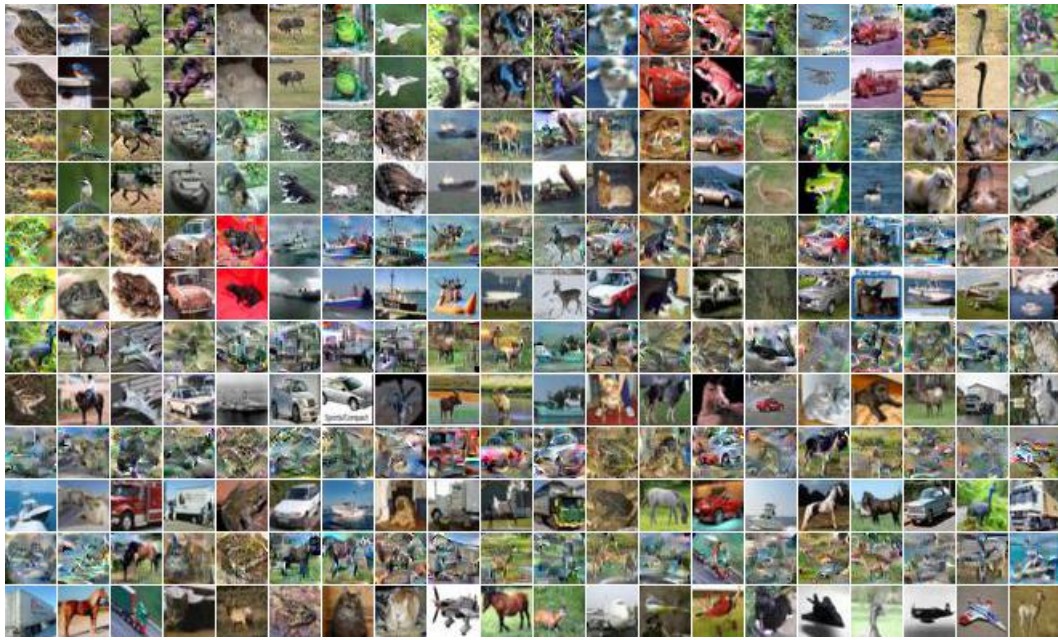

Figure 40: Reconstructions from ResNet trained on 120 images using SimuDy, SSIM = 0.1197.

| Dataset Size | GPU Memory | Reconstruction Time |
|:---:|:---:|:---:|
| 20 | 4334 MB | 1.06 h |
| 30 | 6852 MB | 3.32 h |
| 40 | 8170 MB | 4.67 h |
| 50 | 10518 MB | 6.07 h |
| 60 | 12668 MB | 8.46 h |
| 70 | 14002 MB | 11.37 h |
| 80 | 15728 MB | 12.77 h |
| 90 | 16814 MB | 12.92 h |
| 100 | 19016 MB | 13.70 h |
| 120 | 22272 MB | 15.39 h |

Table 3: GPU memory usage and reconstruction time for training CIFAR-10 datasets with different sizes on one RTX 4090 GPU.

