# OpenReview forum: "Simulating Training Dynamics to Reconstruct Training Data from Deep Neural Networks"
_ICLR.cc/2025/Conference — ICLR 2025 Poster_

### Official Review · Reviewer_NSaC · 2024-10-31

**Soundness:** 3
**Presentation:** 3
**Contribution:** 2
**Rating:** 5
**Confidence:** 3

**Summary:**

This submission proposes a method to reconstruct training data from neural network models by simulating their non-linear training dynamics. It adopts a nested optimization approach, where the outer optimization tunes the data being reconstructed, and the inner optimization performs regular training to match the original model’s parameters. Experimental results demonstrate successful dataset reconstruction for larger, more complex models like ResNet, where previous methods in linear regimes struggle, albeit assuming full access to weights, architecture, and hyperparameters.

**Strengths:**

- The paper is generally well-written and clear, making the core ideas easily accessible.
- The proposed nested optimization approach is a meaningful extension over prior work for the task, effectively addressing non-linear training dynamics in a way that previous methods in linear regimes could not. This method is well-motivated and aligns with the challenges of reconstructing data from more complex models like ResNet.
- Experimental evaluation is thorough and supports the claims within the chosen setting. The authors evaluate across different dataset and model sizes, confirming the performance benefits of SimuDy over baseline methods in similar constrained scenarios.

**Weaknesses:**

- The submission lacks a defined threat model for data reconstruction. That makes it challenging to evaluate the setting, how broad or narrow it is, and what comparable methods might be. It seems to require knowledge about model architecture, weight (history) and hyperparameters - just not the gradients and training data. To me, that seems to be rather specific and I would ask the authors to elaborate on why they chose that particular setting.
- Relatedly, the model-to-data ratio used here raises questions about both the threat model and the generalizability of the findings. A ResNet-18 trained on 120 images represents a highly constrained regime compared to standard machine learning applications. Is model depth less of a challenge here than dataset size? This also leads me to question whether memorization and an overfitted model-to-data ratio are necessary conditions for SimuDy’s success. If not, I would be interested in an experiment using a similarly complex model trained on a larger dataset, such as the full CIFAR-10 or CIFAR-100, with an attempt to reconstruct samples. Even if complete recovery is not feasible, partial reconstruction of a meaningful subset could provide insights into the general applicability of this method.
- Finally, I would like to see a more comprehensive evaluation against baseline methods. Model inversion attacks seem relevant here; if not, I would suggest the authors clarify why these are not comparable and discuss performance and computational cost of either approahces. Such a comparison would contextualize the efficiency and effectiveness of SimuDy against established approaches.

**Questions:**

See weaknesses

---

> ### Author Response · Authors · 2024-11-22
> **Response to Reviewer NSaC (1/2)**
>
> Thank you for your appreciation of our performance and clarity, along with your insightful suggestions. We address your concerns as below:
>
> **R4.1 Setting Definition.**
>
> Thank you for your careful reading and valuable suggestions. We will elaborate on the setting for dataset reconstruction.
>
> We have access to only the initial and final parameters of the model. And, we manage to reconstruct training dataset from the two sets of parameters.
>
> We noticed that you might have a slight misunderstanding regarding our method: you mentioned that we require full access to hyper-parameters. Having access to the training hyper-parameters will help reconstruction, but for more common situations where hyper-parameters are unknown, we have designed a strategy to find suitable hyper-parameters for simulation of dynamics. Specifically, we preset the batch size and tune learning rate by grid search based on the first dozens of steps’ loss. Thus, suitable hyper-parameters can be identified effectively and efficiently. The strategy is introduced in Section 3, and the reconstruction results of different hyper-parameters identified using our strategy are shown in Section 4.3 Fig.7.
>
> The two sets of parameters of the model are accessible in many realistic scenarios, such as fine-tuning from public models, or in federated learning where clients receive period updates of the model parameters, so we choose such setting. In those scenarios, parameters of model are published alongside model architecture, and only publishing parameters is meaningless without the architecture. In short, our setting is to reconstruct training dataset from only the initial and final parameters.
>
> **R4.2 Model Depth, Model-to-Data Ratio & Dataset Size.**
>
> - Challenges of Model Depth
>
> The model depth and dataset size both pose great challenges to dataset reconstruction from the parameters. We first discuss the challenges of model depth. Previous works mainly focus on simple and shallow MLPs, which possess linear training dynamics. However, for deeper and more complex DNNs, the training dynamics become non-linear. We propose to simulate such non-linear dynamics for more challenging and practical reconstruction, which constitutes our main contribution.
>
> - Model-to-Data Ratio
>
> Your question about the overfitted model-to-data ratio is quite insightful. All deep models have sufficient parameters, and a very high model-to-data ratio (in other words, over-parameterization) is necessary for benign overfitting [1, 2]: a model fits the training data well while also generalizing effectively to test data. Memorization also needs a sufficient number of parameters. Therefore, we believe that a high model-to-data ratio is the nature of deep learning and meanwhile is necessary for training data reconstruction.
>
> - Challenges of Dataset Size
>
> Dataset size makes reconstruction challenging primarily from two aspects. Firstly, the larger dataset means more GPU memory usage in simulating the training dynamcis and back-propagating with multiple training steps. Secondly, as the dataset size increases, the uncertainty in the optimization problem of decoupling gradients of more data increases, making reconstruction much more challenging. We mention a similar and easier task called gradient inversion attack, where attackers reconstruct training data from one single gradient. Even without considering decoupling gradients from parameters across different steps, the state-of-the-art methods [3, 4] can reconstruct only about 100 images. The decoupling problem is quite hard to optimize. And we can achieve comparable number of reconstruction images even in the more difficult task.
>
> Thank you for your suggestions to reconstruct a subset of training dataset. The idea is very valuable and meaningful. However, as we mentioned above, the more intensive memory requirements and harder optimization of the decoupling problem make it much more challenging. We attempted to optimize only dozens of dummy images to represent a meaningful subset of full dataset. However, the dummy images became a fusion of multiple original images, similar to the phenomenon observed in Section 4.1, making the reconstructed images appear meaningless. During the rebuttal period, we couldn't come up with a better method to reconstruct a subset of the full dataset. Your idea is very meaningful for deep learning and it deserves further investigation in the future. We also believe that more effective solutions of the optimization problem and efficient simulation for less memory requirements will help solve larger datasets in the future.

---

> > ### Author Response · Authors · 2024-11-22
> > **Response to Reviewer NSaC (2/2)**
> >
> > **R4.3 Relationship with Model Inversion Attack.**
> >
> > In this paper, we compared SimuDy with three pioneering works on the task of training dataset reconstruction from parameters. Since we have overcome the challenges posed by non-linear dynamics, significant performance improvements can be expected.
> >
> > We hope to find more baselines, but falied. Thank you for your valuable suggestions about model inversion (MI) attack. We are not experts in this area, and following your suggestion, we quickly learn some representative works [5, 6, 7, 8, 9, 10, 11], especially a recent survey [12]. If we understand correctly, the MI attack is fundamentally different to our tasks.
> >
> > - MI attacks aim to find a single (typical or individual) sample with similar features or outputs, or to estimate key properties of such a sample. However, our task is different: we aim to reconstruct all training samples from the parameters of a DNN.
> >
> > - MI attacks need the targeted model output, or logits, or features of the sensitive data as the input. And, our task uses DNN's parameters as the input. Generally, MI is to align the forward calculation of a sensitive input and some reference; while ours is to align training trajectory by leveraging the synergy of backward calculations.
> >
> > We will add the discussions and include the mentioned literatures in the final version. Thanks for mentioning MI attacks, and if there are other relevant papers or topics you could bring us, we would greatly appreciate that.
> >
> > [1] Zhang C, Bengio S, Hardt M, et al. Understanding deep learning (still) requires rethinking generalization. ICLR, 2017.
> >
> > [2] Neyshabur B, Li Z, Bhojanapalli S, et al. The role of over-parametrization in generalization of neural networks. ICLR, 2019.
> >
> > [3] Yin H, Mallya A, Vahdat A, et al. See through gradients: Image batch recovery via gradinversion. CVPR, 2021.
> >
> > [4] Zhang C, Xiaoman Z, Sotthiwat E, et al. Generative gradient inversion via over-parameterized networks in federated learning. CVPR, 2023.
> >
> > [5] Fredrikson M, Jha S, Ristenpart T. Model inversion attacks that exploit confidence information and basic countermeasures. ACM CCS, 2015.
> >
> > [6] Hitaj B, Ateniese G, Perez-Cruz F. Deep models under the GAN: information leakage from collaborative deep learning. ACM CCS, 2017.
> >
> > [7] Song C, Ristenpart T, Shmatikov V. Machine learning models that remember too much. ACM CCS, 2017.
> >
> > [8] Melis L, Song C, De Cristofaro E, et al. Exploiting unintended feature leakage in collaborative learning. IEEE S&P, 2019.
> >
> > [9] Yang Z, Zhang J, Chang E C, et al. Neural network inversion in adversarial setting via background knowledge alignment. ACM CCS, 2019.
> >
> > [10] He Z, Zhang T, Lee R B. Model inversion attacks against collaborative inference. ACM ACSAC, 2019.
> >
> > [11] Wang Z, Song M, Zhang Z, et al. Beyond inferring class representatives: User-level privacy leakage from federated learning. IEEE INFOCOM, 2019.
> >
> > [12] Dibbo S V. Sok: Model inversion attack landscape: Taxonomy, challenges, and future roadmap. IEEE CSF, 2023.

---

> > > ### Comment · Reviewer_NSaC · 2024-11-22
> > >
> > > I thank the authors for their response. I appreciate the discussion of MI, and accept that direct comparisons may be challenging.
> > >
> > > In my view, the other two points still stand. The threat model should be made explicit. I think it's fine to choose particular settings, like having access to initial and final parameters, and the exact number of training samples used, potentially some hyper-parameter (ranges), but I would encourage the authors to discuss why they chose this and how working on this particular setting helps towards more general settings.
> > >
> > > Similarly, I remain skeptical on the model-to-data ratio and impact of potential overfitting. I agree with the authors that overparameterization is common. That said, the ratio of ResNet-18-sized models to 10s to 100-s samples is on the extreme end of the spectrum. Whether or not overfitting plays a role is hard to say, since the submission does not state the number of epochs used for fine-tuning, nor provide any performance metrics that could help disentangle fine-tuning performance / overfitting from reconstruction performance. Again, assuming overfitting - if necessary - may be part of the threat model, but it should be explicit.
> > >
> > > As is, I find it challenging to identify what specifically one can take away from the submission. Under certain conditions, reconstructing small number of training samples from large models is possible - which indicates memorization. What conditions, where it works and where it does not, remains unclear to me. For that reason, I keep my score.

---

> > > > ### Author Response · Authors · 2024-11-22
> > > >
> > > > First of all, we would like to thank your valuable reply, the related discussions are quite helpful.
> > > >
> > > > Your first question concerns the reason for choosing such a setting. In all previous works [1, 2, 3] in the dataset reconstruction field, the general setting is reconstructing training data from the parameters of trained models. We follow their setting for a comprehensive comparison with their methods and extend reconstruction from MLPs to DNNs. And we have shown the robustness of our method without knowing exact number of training samples in Sec 4.5. If our understanding of the general setting differs from yours, we would greatly appreciate it if you could share your perspective on general setting.
> > > >
> > > > The primary motivation of dataset reconstruction is to investigate memorization of models. The training of DNNs is highly non-convex and thus the final parameters are determined by the data and initial parameters together. In other words, data information is embedded (or memorized) into the change of parameters. Therefore, knowing both the initial and final parameters is necessary for dataset reconstruction setting. And there are such scenarios in real, such as fine-tuning from public models.
> > > >
> > > > Your second question is whether overfitting is necessary for the successful reconstruction. We trained the ResNet-18 on 120 CIFAR-10 images until the training accuracy reaches 100%. The number of epochs is 16 and the training loss and test accuracy on CIFAR-10 test set of 10000 images are shown below.
> > > >
> > > > | epoch | 1 | 2 | 3 | 4 | 5 | 6 | 7 | 8 | 9 | 10 | 11 | 12 | 13 | 14 | 15 | 16 |
> > > > | ---| --- | --- | --- | --- | --- | --- | --- | --- | --- | --- | --- | --- | --- | --- | --- | --- |
> > > > | train_loss | 7.03 | 6.76 | 6.53 | 6.29 | 6.03 | 5.71 | 5.29 | 4.76 | 4.03 | 3.17 | 2.44 | 1.73 | 1.15 | 0.80 | 0.43 | 0.28 |
> > > > | test_acc(%) | 8.64 | 10.87 | 14.12 | 17.97 | 24.18 | 31.26 | 36.78 | 41.25 | 44.26 | 47.34 | 48.76 | 50.17 | 48.54 | 51.13 | 52.8 | 54.49 |
> > > >
> > > > One can see that the test accuracy increases as training loss decreases, showing such training process is not overfitting. Your suggestion on showing the performance metrics of training is quite meaningful, we will certainly include it in the final version. To reconstruct the training dataset, sufficient training to feed sufficient information into the model is necessary. Overfitting receives even more gradients of data and will also lead to successful reconstruction, but it is not necessary for our success.
> > > >
> > > > Overall, thanks for discussing the conditions of the successful reconstruction. Our answer is for an over-parameterization model, when it is well trained (without requiring overfitting), it is possible to reconstruct the training dataset from known initial and final parameters.
> > > >
> > > > [1] Haim N, Vardi G, Yehudai G, et al. Reconstructing training data from trained neural networks. NeurIPS, 2022.
> > > >
> > > > [2] Buzaglo G, Haim N, Yehudai G, et al. Deconstructing data reconstruction: Multiclass, weight decay and general losses. NeurIPS, 2024.
> > > >
> > > > [3] Loo N, Hasani R, Lechner M, et al. Understanding Reconstruction Attacks with the Neural Tangent Kernel and Dataset Distillation. ICLR, 2024.

---

> > > > > ### Author Response · Authors · 2024-11-28
> > > > >
> > > > > The previous discussion with you on overfitting is quite interesting. We look forward to further valuable suggestions from you. We also believe including some conclusions from our discussion in the paper could benefit the readers.
> > > > >
> > > > > The deadline of the PDF revision is approaching, we have to update a version to respond to other reviewers. We feel sorry for not including the discussion on overfitting at this stage, but we look forward to incorporating the consensus with you in the final version of the PDF.

---

### Official Review · Reviewer_v7cX · 2024-11-02

**Soundness:** 3
**Presentation:** 3
**Contribution:** 2
**Rating:** 6
**Confidence:** 3

**Summary:**

The paper proposes Simulation of training Dynamics (SimuDy), an easy and effective approach that reconstructs datasets from neural networks by optimizing dummy datasets through the comparison of resulting network parameters after training. The proposed method demonstrates better performance compared to existing baselines and shows promising results on relatively more advanced architectures like ResNets.

**Strengths:**

1. The proposed method has much better performance compared to existing baseline. The method is also easy to implement.
2. This work is both novel and interesting to me.

**Weaknesses:**

I am mainly concerned about the implication and the usefulness of the proposed method.
1. **Practicality and Scalability.** Back propagating with multiple training steps can be computationally intensive and challenging to scale for larger models and with larger datasets with high resolutions. Additionally, the method requires prior knowledge of hyper-parameters such as the number of training steps, batch size, and weights at initialization/end which limits its applicability.
2. **Implication and Usefulness.** I wonder how SimuDy can contribute to a deeper understanding of neural network training dynamics. It difficult to assess its broader usefulness in the field of machine learning.

**Questions:**

See weaknesses.

---

> ### Author Response · Authors · 2024-11-22
> **Response to Reviewer v7cX**
>
> Thank you for your appreciation of performance and novelty of SimuDy. We address your concerns as below:
>
> **R3.1 Practicality & Scalability.**
>
> The task of dataset reconstruction from parameters of DNNs is quite challenging. Compared to previous works [1, 2, 3] in dataset reconstruction from parameters, we extend dataset reconstruction from shallow and small MLPs to more pratical DNNs. Moreover, our method SimuDy can generalize well to larger models and higher-resolution datasets, with qualitative recosntruction results of a Vision Transformer trained on ImageNet shown in Appendix C.5 Fig.19.
>
> We propose to simulate non-linear training dynamics, and as you mentioned, back propagating with multiple training steps is computationally intensive and thus makes scale for larger-size datasets challenging. For a similar but simpler task called the gradient inversion attack (GIA), where attackers reconstruct images from a single gradient, state-of-the-art methods [4, 5] can reconstruct approximately 100 images. We achieve a comparable number of reconstructed images, even for this more challenging task. Moreover, our primary motivation is to verify the memorization of DNNs, rather than to attack DNNs in practice. We believe that our methods would provide valuable insights into the memorization of DNNs.
>
> We noticed that there might be a slight misunderstanding regrading our proposed method. Provided only the initial and final parameters, SimuDy can reconstruct training data. Prior knowledge of hyper-parameters, such as the number of training steps, batch size and learning rate, is not necessary for reconstruction. We have developed a strategy to find suitable hyper-parameters for simulating training dynamics, with the details in Section 3 and experimental results in Section 4.3.
>
> [1] Haim N, Vardi G, Yehudai G, et al. Reconstructing training data from trained neural networks. NeurIPS, 2022.
>
> [2] Buzaglo G, Haim N, Yehudai G, et al. Deconstructing data reconstruction: Multiclass, weight decay and general losses. NeurIPS, 2024.
>
> [3] Loo N, Hasani R, Lechner M, et al. Understanding Reconstruction Attacks with the Neural Tangent Kernel and Dataset Distillation. ICLR, 2024.
>
> [4] Yin H, Mallya A, Vahdat A, et al. See through gradients: Image batch recovery via gradinversion. CVPR, 2021.
>
> [5] Zhang C, Xiaoman Z, Sotthiwat E, et al. Generative gradient inversion via over-parameterized networks in federated learning. CVPR, 2023.
>
> **R3.2 Implication & Usefulness.**
>
> We analyze the linearity of training dynamics for MLPs and ResNets, as shown in Appendix A. We also design a metric to measure the linearity. We consider simulating non-linear training dynamics as a tool for reconstruction from parameters of DNNs. The success of SimuDy indicates that the parameters and the data are intrinsically linked through the training dynamics. Moreover, a better understanding of training dynamics will aid reconstruction, and it would be interesting to explore these dynamics further, such as leveraging the low-dimensional nature of the training process [1, 2] to enable efficient simulation.
>
> Deep learning has achieved marvelous success. However, the mechanism, especially why over-parameterized DNNs can generalize quite well to unseen data, is still unclear. Memorization is closely related to generalization [3] and raises concerns about the potential leakage of private information within training data.
>
> Therefore, verifying or falsifying the presence of memorization in deep models is crucial. Recently, there have been many discussions on membership inference attacks (MIA) [4], which aim to determine whether a specific data point was included in the training dataset. MIA serves as an indicator of whether the model has memorized the data point.
>
> More direct evidence is to reconstruct the training set from a model's parameters. Therefore, pioneering works in this field have attracted significant attention. Our contribution is that we have overcome the limitations of the linearity of the training process. This step is quite significant since real-world training is certainly non-linear. SimuDy can then be used to extract memorization from ResNet, as well as ViT and tinyBERT. We think SimuDy can indeed confirm that DNNs can remember some training samples, which may lead to further exploration into the relationship between memorization and generalization, interpretability, as well as security and data privacy.
>
> [1] Gur-Ari G, Roberts D A, Dyer E. Gradient descent happens in a tiny subspace. arXiv, 2018.
>
> [2] Li T, Tan L, Huang Z, et al. Low dimensional trajectory hypothesis is true: Dnns can be trained in tiny subspaces. TPAMI, 2022.
>
> [3] Feldman V, Zhang C. What neural networks memorize and why: Discovering the long tail via influence estimation. NeurIPS, 2021.
>
> [4] Shokri R, Stronati M, Song C, et al. Membership inference attacks against machine learning models. IEEE S&P, 2017.

---

> > ### Comment · Reviewer_v7cX · 2024-11-26
> >
> > Thank you for your responses. I have decided to keep my original rating.

---

> > > ### Author Response · Authors · 2024-11-28
> > >
> > > Thank you sincerely for the meaningful and valuable discussion which helps improve our work.

---

### Official Review · Reviewer_aded · 2024-11-03

**Soundness:** 3
**Presentation:** 3
**Contribution:** 3
**Rating:** 8
**Confidence:** 3

**Summary:**

This paper addresses the data reconstruction problem for more realistic neural networks (particularly ResNet-18) than previous works. They propose to optimize the input data so that the training direction by the optimized data aligns with the direction trained by the target data. The experiments show that the proposed method successfully reconstruct CIFAR10 data from the trained parameter (and also the initial parameter) of ResNet-18. The method and results are really fascinating, but there are some concerns in their experiments as explained below.

**Strengths:**

1. The paper is clearly written and easy to follow. The method is also clearly described.
2. The proposed method is derived in a straightforward way, and the implementation is also very simple. In spite of these simplicity, the results seem really impressive compared to previous works. I like both the idea and the simplicity.

**Weaknesses:**

1. The method assumes the initialization parameter to be given a priori, which is an unrealistic assumption for data reconstruction attack. However, I think this is not a critical issue because the contributions of this paper may not be limited to such attacks.
2. As authors also mentioned, the computational and memory budget may be terribly large due to higher order differentiation. Although this does not hurt the contribution of this work, it should be worth reporting the actual reconstruction time.
3. In experiments, the proposed method is only compared with Loo et al.'s method [1]. However, it should be also (rather) compared with Buzaglo et al.'s [2] (or Haim et al. [3]) method since this is the first successful method for data reconstruction from parameters. Moreover, to verify the generality of the proposed method, i.e., to check the results being not overfitting to CIFAR10 with ResNet18, the experiments should include several datasets (e.g. SVHN or MNIST) and model architectures like other ResNet architectures etc, with quantitative comparison to the baselines.
4. It is unclear how the coefficient of the TV regularization affects the quality of reconstructed data. It should be worth reporting the quantitative analysis from this perspective.
5. There are some missing related works [4,5] on gradient/trajectory matching for non-linear training dynamics. Zhao et al. [4] optimizes the input space by the cosine symmetry of two training dynamics, and hence is strongly related to this paper. Chijiwa [5] also optimizes the cosine similarity of two training dynamics, but focusing on the symmetry in a neural network rather than input data.

[1] Loo et al. "Understanding Reconstruction Attacks with the Neural Tangent Kernel and Dataset Distillation" (ICLR'24)

[2] Buzaglo et al. "Deconstructing Data Reconstruction: Multiclass, Weight Decay and General Losses" (NeurIPS'23)

[3] Haim et al. "Reconstructing Training Data from Trained Neural Networks" (NeurIPS'22)

[4] Zhao et al., "Dataset Condensation with Gradient Matching" (ICLR'21)

[5] Chijiwa, "Transferring Learning Trajectories of Neural Networks" (ICLR'24)

**Questions:**

See weaknesses.

---

> ### Author Response · Authors · 2024-11-22
> **Response to Reviewer aded (1/2)**
>
> Thank you for your appreciation of our performance and clarity, along with your insightful suggestions. We address your concerns as below:
>
> **R2.1 Known Initilization Parameters.**
>
> Thank you for recognizing that access to initial parameters is not a critical issue. Our main motivation and contribution is not to attack DNNs in practice but to verify the memorization of DNNs. SimuDy is more like an alarm for data security, rather than a practical threat.
>
> We agree with you that knowing the initial parameters is unrealistic in some scenarios. However, access to initial parameters arises naturally in many settings, such as fine-tuning from public models or in federated learning where clients receive period updates of the model parameters.
>
> **R2.2 Computational and Memory Budget.**
>
> Thank you for your understanding. Following your suggestion, we have reported the reconstruction time and GPU memory usage of SimuDy on CIFAR-10 for different-sized datasets in Appendix C.7 Tab.3, also shown following. Notice that for reconstruction of 120 images, we almost run out of the GPU memory on an RTX 4090 (24GB).
>
> | Dataset Size | GPU Memory | Reconstruction Time |
> | -------- | -------------- | ------------- |
> | 20 | 4334 MB | 1.06 h |
> | 30 | 6852 MB | 3.32 h |
> | 40 | 8170 MB | 4.67 h |
> | 50 | 10518 MB | 6.07 h |
> | 60 | 12668 MB | 8.46 h |
> | 70 | 14002 MB | 11.37 h |
> | 80 | 15728 MB | 12.77 h |
> | 90 | 16814 MB | 12.92 h |
> | 100 | 19016 MB | 13.70 h |
> | 120 | 22272 MB | 15.39 h |
>
> **R2.3.1 Comparison with Pioneering Works.**
>
> Thank you for your valuable suggestions. Haim et al. [1] first show that the parameters of trained networks contain information to reconstruct training samples. They successfully reconstruct a dataset from a binary MLP. Buzaglo et al. [2] extend reconstruction to a multi-class setting. Both of these pioneering works inspire us greatly. We take a step toward reconstructing training data from the parameters of more complex DNNs. The essential progress is that we can reconstruct data from non-linear training dynamics, whereas previous works are effective only for linear dynamics.
>
> The qualitative and quantitative reconstruction results of Buzaglo et al.'s method have been updated in Appendix C.4 since our method mainly focuses on the multi-class setting. When applied to ResNet-18 with non-linear training dynamics, Buzaglo et al. use only the gradients of the final models to characterize the entire dynamics, which is inadequate and results in degraded reconstruction quality. Moreover, their method requires very small initial parameters or the application of weight decay during training, and the absence of these factors could further degrade performance.
>
> [1] Haim N, Vardi G, Yehudai G, et al. Reconstructing training data from trained neural networks. NeurIPS, 2022.
>
> [2] Buzaglo G, Haim N, Yehudai G, et al. Deconstructing data reconstruction: Multiclass, weight decay and general losses. NeurIPS, 2024.
>
> **R2.3.2 Generality of SimuDy on Different Datasets and Model architectures.**
>
> Thank you for your valuable suggestions on verifying the generality of SimuDy. We had previously included experiments on another dataset, Tiny ImageNet, and the results can be found in Appendix C.1, Fig. 12. To further verify the generality of SimuDy, we conducted new experiments:
>
> - SVHN dataset and ResNet-50 architecture: The reconstruction results have been reported in Appendix C.2, Fig.13 and Fig.14.
>
> - ViT trained on ImageNet : The reconstruction results have been reported in Appendix C.5, Fig.19.
>
> - TinyBERT trained on CoLA in NLP: The reconstruction results have been reported Appendix C.6, Tab.2.
>
> All the above experiments show that SimuDy can generalize well to a variety of datasets and model architectures.
>
> **R2.4 The Affect of Coefficient of TV Loss.**
>
> To answer your question, we discretely selected the coefficient $\alpha$ of the TV loss. Both the qualitative and quantitative reconstruction results can be seen in Appendix C.3.
>
> As the coefficient $\alpha$ increases, the quantitative SSIM value initially improves, reaching a peak, and then decreases thereafter. Thus, selecting an appropriate coefficient is crucial, as both excessively large and small values negatively affect reconstruction quality. Interestingly, an excessively large coefficient can cause reconstructed images to become blurry, even resulting in color shifts compared to the original training images, as shown in Appendix C.3 Fig. 17.

---

> > ### Author Response · Authors · 2024-11-22
> > **Response to Reviewer aded (2/2)**
> >
> > **R2.5 Discussion about Related Works.**
> >
> > Thank you for your additions to the related works of our paper. Both works are related to training dynamics and cosine similarity.
> >
> > Zhao et al. [1] minimize gradient matching loss between minibatch of synthetic set and that of original dataset. Their method has similarities with ours, but is essentially different. First of all, we have different aims. They compress the accessible dataset into a smaller sample set for more efficient training later, whereas we manage to reconstruct the whole training dataset from only trained model parameters without any prior knowledge of the original dataset. Secondly, they align the gradients of minibatches, while we align the parameter differences, i.e. we align the whole training dynamics. They align the gradients of minibatches across different model parameters, rather than simulating the true training dynamics.
> >
> > Chijiwa [2] transfers a known training trajectory from the source initialization to the target initialization and generates a new training trajectory, allowing the new model to achieve non-trivial accuracy prior to additional training and significantly speeding up the subsequent training. Chijiwa also uses cosine similarity for optimization. The learning transfer problem is novel and we believe it will help us further understand training dynamics.
> >
> > The two related works and the discussion will be added in the final version.
> >
> > [1] Zhao B, Mopuri K R, Bilen H. Dataset condensation with gradient matching. ICLR, 2021.
> >
> > [2] Chijiwa D. Transferring learning trajectories of neural networks. ICLR, 2024.

---

> > > ### Comment · Reviewer_aded · 2024-11-24
> > >
> > > Thank you for additional clarification and experiments. Based on these consistent results, I raised my score. However,
> > >
> > > > The qualitative and quantitative reconstruction results of Buzaglo et al.'s method have been updated in Appendix C.4 since our method mainly focuses on the multi-class setting.
> > >
> > > I think this comparison should be done in the experiments of main text, instead of the isolated result in Appendix. Also, the multi-class setting is not a valid reason for avoiding such comparison since the proposed method can be also applied to binary classification setting.
> > >
> > > In addition, the results in Appendix C.7 should be placed side by side for each method, so that the reader can easily compare the proposed method and baselines.

---

> > > > ### Author Response · Authors · 2024-11-25
> > > >
> > > > Thank you for your appreciation of our clarification and experiments. We would also like to express our sincerest thanks for your time in helping improve our work.
> > > >
> > > > Indeed, it is better to report Appendix C.4 in the main text. For space saving, we plan to merge the comparison with Buzaglo et al.'s method in Fig. 5, together with other methods. Fig. 5 corresponds to the reconstruction of 50 images, whereas Appendix C.4 focuses on a 20-image reconstruction (and the reconstruction performance is already not good). We are afraid that the performance of Buzaglo et al.'s method on 50-images reconstruction might be poor and lose the value of comparison in our setting. If in this case, we would prefer to report the comparision on the 20-image task in the appendix; if not, certainly we will report the 50-image task in Fig. 5 for a comprehensive comparison.
> > > >
> > > > We totally agree with your comments on binary classification setting. SimuDy certainly can be used for binary classification. And we will ensure to include comparisons of binary setting, along with results of different-size datasets placed side by side for each method in Appendix C.7 of the final version as soon as the additional experiments have been completed.

---

> > > > > ### Author Response · Authors · 2024-11-28
> > > > >
> > > > > Following your valuable suggestions, we have updated the comparison with Buzaglo et al.'s method in Section 4 of the main text. Reconstructions of binary classification setting have been updated in Appendix C.5. Moreover, the reconstruction results of different-size datasets have been placed side by side for each method in Appendix C.8.
> > > > >
> > > > > We would like to express our sincerest thanks again to you for your efforts in improving and re-evaluating our work.

---

### Official Review · Reviewer_KK4w · 2024-11-03

**Soundness:** 3
**Presentation:** 3
**Contribution:** 3
**Rating:** 6
**Confidence:** 3

**Summary:**

SimuDy, a novel framework designed to reconstruct training data from the parameters of deep neural networks. SimuDy simulates the complex, non-linear training dynamics characteristic of deep architectures like ResNet by optimizing a dummy dataset to match the final parameters of a trained model. The authors demonstrate that SimuDy can accurately recover training samples under various conditions, including scenarios with unknown hyperparameters, outperforming previous linear-dynamics-based methods.

**Strengths:**

1.  Novelty: By simulating non-linear dynamics, SimuDy extends dataset reconstruction capabilities to deeper models, such as ResNet, making it applicable to practical, real-world architectures rather than limiting it to simpler models.
2. Robustness to Unknown Hyperparameters: SimuDy’s ability to reconstruct data even without exact training hyperparameters is a notable contribution, as real-world scenarios often lack complete information about training configurations.
3. Quantitative and Qualitative Performance Improvements: SimuDy outperforms existing methods both in terms of structural similarity index and visual quality of reconstructed samples.
4. The authors conduct extensive experiments, including robustness tests across varying dataset sizes, initializations, and dummy dataset configurations.

**Weaknesses:**

1. While SimuDy demonstrates promising results in reconstructing training samples from model parameters, there is limited discussion on the practical implications such as privacy or data security or in federated learning.
2. SimuDy requires storing the entire training computation graph, which may limit its feasibility for very large models or datasets. The paper would benefit from a detailed breakdown of computational costs and memory requirements, especially for scaling to industry-sized datasets.
3. The initialization of dummy datasets appears somewhat arbitrary (e.g., using noise or natural images). Conduct a systematic study on different initialization techniques for dummy datasets, such as starting with more structured priors, to assess their impact on reconstruction quality and convergence speed.
4. Although SimuDy achieves high-quality reconstructions for small datasets, it’s unclear if these results generalize to larger, more diverse datasets. Extend experiments to larger datasets like ImageNet or tasks outside image classification to validate the scalability and generalization of SimuDy. Also it will be interesting to see the results of reconstruction when multiple objects are involved in an image.
5. Lack of experiments in Vision Transformers. It will be interesting to extend some experiments on Vision Transformers, which differ from CNNs in their patch-based processing and self-attention mechanisms. Also, perform a comparative study to examine the differences in reconstruction quality between CNNs and transformers.

**Questions:**

See the weakness section.

---

> ### Author Response · Authors · 2024-11-22
> **Response to Reviewer KK4w (1/2)**
>
> Thank you for your appreciation of the novelty of SimuDy and insightful suggestions. We address your concerns as below:
>
> **R1.1 Privacy & Security.**
>
> We sincerely appreciate your recognition of SimuDy's performance. Reconstruction from parameters is a direct evidence that DNNs remember something through training. This could reveal the mechanism of deep learning and meanwhile pose challenges for privacy and data security, as you concerned.
>
> Pioneering works rely on linear training dynamics. We think stepping over the gap between linear and non-linear dynamics is quite essential towards the real-world DNNs, which is the motivation of our work. Frankly speaking, we did not anticipate that the reconstruction results would be so impressive, both quantitatively and qualitatively, even without precise knowledge of the training hyper-parameters.
>
> Indeed, towards practical implications, there are still many aspects that warrant further consideration and resolution, including the number of training samples, the significant computational costs, and the substantial memory requirements. We hope SimuDy's successful reconstruction could open a door towards the memorization of real-world DNNs and serve as an alarm on data security, rather than an immediate threat.
>
> **R1.2 Computational Costs & Memory Requirements.**
>
> Indeed, the computational costs and memory requirements of SimuDy are currently heavy. This is because dataset reconstruction from DNNs with non-linear training dynamics is quite difficult and requires storing the entire training computation graph, as you mentioned.
>
> We ourselves regard SimuDy as the first step to extract memory from real-world DNNs, rather than focusing on immediate attack performance.
>
> And we do agree with the reviewer that it will be interesting and meaningful to explore methods of reducing computational resource demands to make SimuDy more practical. We believe that SimuDy will benefit from that, especially considering that attackers are not likely to be frugal with computational resources.
>
> **R1.3 Initialization of Dummy Datasets.**
>
> According to your valuable suggestion, besides random noise and natural images, we add two initializations that have more priors: one is the original training images with added noise, and the other is original images where the central 16x16 pixels are set to white. The order of reconstruction quality is as follows: original images with added noise > noise > original images with central 16x16 pixels white > other natural images. The convergence speed follows the same order.
>
> The initializations with priors should help better reconstruction and speed up convergence. But original images with central 16x16 pixels white resulted in worse performance than using noise as the initialization. This indicates that the similarity in the view of humans and DNNs is not the same, which makes prior design quite hard.
>
> Prior knowledge of the original dataset is typically unavailable, making it reasonable to use noise as initialization. Most data recovery methods [1, 2, 3, 4] based on optimization use random noise as initialization. SimuDy can successfully reconstruct training data from different initializations.
>
> [1] Zhu L, Liu Z, Han S. Deep leakage from gradients. NeurIPS, 2019.
>
> [2] Geiping J, Bauermeister H, Dröge H, et al. Inverting gradients-how easy is it to break privacy in federated learning? NeurIPS, 2020.
>
> [3] Haim N, Vardi G, Yehudai G, et al. Reconstructing training data from trained neural networks. NeurIPS, 2022.
>
> [4] Loo N, Hasani R, Lechner M, et al. Understanding Reconstruction Attacks with the Neural Tangent Kernel and Dataset Distillation. ICLR, 2024.

---

> > ### Author Response · Authors · 2024-11-22
> > **Response to Reviewer KK4w (2/2)**
> >
> > **R1.4 & R1.5 Reconstructions in NLP Regime & in ViTs with ImageNet.**
> >
> > Thanks for your meaningful suggestion on extending SimuDy to different tasks. We guess memorization happens in most deep models. The previous works on reconstruction from parameters only consider image classification for NNs, but we are happy to conduct additional experiments: reconstruction in NLP regime and ViT trained on ImageNet.
> >
> > - Reconstruction in NLP regime.
> >
> > We choose TinyBERT as the model and CoLA as the dataset. Experiments are conducted on one RTX 4090 GPU. In the SOTA gradient inversion attack methods in NLP [1], a size-of-4 batch of data is recovered from the gradient of TinyBERT. Reconstruction from parameters is much more challenging, we train the model and apply SimuDy to parameters of TinyBERT. We show part of the reconstructed sentences below: the left column is the original samples fed into the model, the middle column is the initial dummy sentences, and the right column is the reconstructed results.
> >
> >
> > | Original | Initialization  | Reconstructed |
> > | -------- | -------- | -------- |
> > | [CLS] who do you think that will question seamus first? [SEP] | [CLS]quistanger fixingimeter cpc forbidden nehru tread terminology [SEP] | [CLS] who do you think will do first question seamus? [SEP] |
> > | [CLS] the boy ran. [SEP] | [CLS] [PAD] [PAD] [SEP] [PAD] foo nightmares [PAD] [PAD] 102 drawer [PAD] | [CLS] the boy ran. [SEP] |
> > | [CLS] i wonder who bill saw and liked mary. [SEP] | [CLS] essays carltonomy crestedhiskhandnac [SEP] vita gail [PAD] | [CLS] i saw mary wonder who liked bill anything. [SEP] |
> > | [CLS] harriet alternated folk songs and pop songs together. [SEP] | [CLS] donetsk dominance grossedlok ass somerset registrar rochdale ins cher [SEP] | [CLS] harriet alternated alternate folk and pop songs together. [SEP] |
> >
> > More reconstructed samples are shown in Appendix C.6. The results show that SimuDy can be used in NLP regime and validate that TinyBERT's parameters indeed remember some sentences.
> >
> > - Reconstruction for ViT with ImageNet.
> >
> > Vision Transformers (ViTs) differ from CNNs, using patching to divide the image and capturing global image features through self-attention. The reconstruction on ViTs would become interesting. For the training dataset, we choose ImageNet with a larger resolution of 224 * 224. To fit the architecture of ViT, we slightly modify SimuDy. This modification is based on an observation presented in APRIL [2]: aligning the gradients of positional embedding effectively helps ensure the correct positioning of patches. We adjust the reconstruction loss for ViTs as following:
> > $$
> > \mathcal{L}\_{\rm recon}(\boldsymbol{x}, \hat{\boldsymbol{\theta}}\_f;\boldsymbol{\theta}\_0,\boldsymbol{\theta}\_f) = - \frac{\langle \boldsymbol{\theta}\_f - \boldsymbol{\theta}\_0,\hat{\boldsymbol{\theta}}\_f - \boldsymbol{\theta}\_0 \rangle}{\|\boldsymbol{\theta}\_f - \boldsymbol{\theta}\_0\|\|\hat{\boldsymbol{\theta}}\_f - \boldsymbol{\theta}\_0\|} - \alpha \cdot \frac{\langle \boldsymbol{\theta}\_{f, E_{pos}} - \boldsymbol{\theta}\_{0, E\_{pos}},\hat{\boldsymbol{\theta}}\_{f, E\_{pos}} - \boldsymbol{\theta}\_{0, E\_{pos}} \rangle}{\|\boldsymbol{\theta}\_{f, E\_{pos}} - \boldsymbol{\theta}\_{0, E\_{pos}}\|\|\hat{\boldsymbol{\theta}}\_{f, E\_{pos}} - \boldsymbol{\theta}\_{0, E\_{pos}}\|} + \beta \cdot {\rm TV} (\boldsymbol{x})
> > $$ Both training and reconstruction run on one RTX 4090, and we successfully reconstruct 10 images from a trained ViT used in [2]. The successful reconstruction validates the scalability of SimuDy to larger images and ViT architectures. The qualitative results have been updated in Appendix C.5. Interestingly, when multiple objects are involved in an image, SimuDy can recover all objects. For example, both a woman and a dog are reconstructed successfully, as shown in the first image of Fig. 19 in Appendix C.5.
> >
> > We also acknowledge your valuable suggestion to enlarge the dataset. However, due to the GPU memory limitation discussed in R1.2, enlarging the dataset becomes much more challenging. The limitation on dataset size is not a specific weakness of SimuDy but rather a common difficulty in the data reconstruction regime. Consider the gradient inversion attack, a much easier task, where attackers reconstruct data from a single gradient. Current state-of-the-art (SOTA) methods [3, 4] can reconstruct about 100 images. At present, the experiments for SimuDy achieve a comparable number of reconstructed images.
> >
> > [1] Balunovic M, Dimitrov D, Jovanović N, et al. Lamp: Extracting text from gradients with language model priors. NeurIPS, 2022.
> >
> > [2] Lu J, Zhang X S, Zhao T, et al. April: Finding the achilles' heel on privacy for vision transformers. CVPR, 2022.
> >
> > [3] Yin H, Mallya A, Vahdat A, et al. See through gradients: Image batch recovery via gradinversion. CVPR, 2021.
> >
> > [4] Zhang C, Xiaoman Z, Sotthiwat E, et al. Generative gradient inversion via over-parameterized networks in federated learning. CVPR, 2023.

---

> > > ### Comment · Reviewer_KK4w · 2024-11-24
> > >
> > > Thank you for addressing the questions. The additional experiments on ImageNet, transformer-based architectures, and NLP tasks enhance the paper's generalizability and relevance. However, a discussion comparing memorization across architectures, such as CNNs and transformers, would be a valuable addition. I found it challenging to interpret the results given differences in datasets and experimental setups.

---

> > > > ### Author Response · Authors · 2024-11-25
> > > >
> > > > Thanks for your recognition of the generalizability and relevance of our work.
> > > >
> > > > Your suggestion on discussing the memorization across architectures is very very important. Memorization happens when a deep model learns the data well, so we believe architectures that have good learning behavior also exhibit better memorization. However, on the other hand, a more efficient architecture generally requires more parameters, denser connections, or deeper layers, all of which imply more complicated learning dynamics, making reconstruction harder.
> > > >
> > > > Thus, we have to admit that we are still at the begining of understanding memorization and cannot provide substantial comments on the role of architectures.
> > > >
> > > > In our experiments, we observed an interesting phenomenon. If we did not apply positional embedding constraints (as explained in our previous response), we could reconstruct images with correct patches but misaligned. By adding the loss term for the positional embedding parameters, we are able to effectively mitigate this issue. This indicates the information of patch position maybe is memorized into parameters of the positional embedding module, showing a potential link between memorization and architectures. However, the evidence is not very sufficient, so we prefer to share this observation here only for discussion.

---

> > > > > ### Comment · Reviewer_KK4w · 2024-11-26
> > > > >
> > > > > Thank you for your clarification and discussion. I have increased my original rating.

---

> > > > > > ### Author Response · Authors · 2024-11-28
> > > > > >
> > > > > > According to your valuable suggestions, we have updated the experiments on ImageNet, transformer-based architectures, and NLP tasks in the latest version of the PDF.
> > > > > >
> > > > > > We are very grateful for your efforts in improving and re-evaluating our work.

---

### Meta-Review · Area_Chair_2E8w · 2024-12-20

**Metareview:**

The paper proposes a method that reconstructs training datasets from the parameters of deep neural networks (DNNs) by simulating non-linear training dynamics. The authors validate SimuDy across multiple tasks, datasets, and architectures, emphasizing the implications for understanding DNN memorization and raising concerns about data privacy. Strengths include its novelty and practical insights into the relationship between model parameters and training data. However, limitations include scalability challenges, reliance on specific assumptions (access to initial and final parameters), and a lack of broader comparative baselines. Overall the reviewers found the strengths outweighs the weakness.

**Additional Comments On Reviewer Discussion:**

While reviewers appreciated the novelty and robustness of SimuDy, concerns were raised about the scalability of the method, the role of overfitting, and the model-to-data ratio. The authors clarified that overfitting is not essential for successful reconstruction and highlighted scenarios where their assumptions (e.g., access to parameters) are realistic. Another concern was the need for a clearly defined threat model.  Reviewers also suggested additional comparisons to model inversion attacks, which the authors argued are fundamentally different from dataset reconstruction. The addition of experiments on NLP tasks and transformer architectures addressed some generalization concerns, leading to improved ratings from certain reviewers. However, skepticism remains about the broader applicability and computational feasibility of SimuDy, particularly for large-scale datasets and real-world settings.

---

### Decision · Program_Chairs · 2025-01-22

Accept (Poster)